# Microbiota functional activity biosensors for characterizing nutrient metabolism in vivo

**Darryl A Wesener[1,2], Zachary W Beller[1,2], Samantha L Peters[3], Amir Rajabi[4], Gianluca Dimartino[4], Richard J Giannone[3], Robert L Hettich[3], Jeffrey I Gordon[1,2]***

[1]Edison Family Center for Genome Sciences and Systems Biology, Washington University School of Medicine, St. Louis, United States; [2]Center for Gut Microbiome and Nutrition Research, Washington University School of Medicine, St. Louis, United States; [3]Chemical Sciences Division, Oak Ridge National Laboratory, Oak Ridge, United States; [4]Mondelez International, Deerfield, United States

**Abstract** Methods for measuring gut microbiota biochemical activities in vivo are needed to characterize its functional states in health and disease. To illustrate one approach, an arabinan-containing polysaccharide was isolated from pea fiber, its structure defined, and forward genetic and proteomic analyses used to compare its effects, versus unfractionated pea fiber and sugar beet arabinan, on a human gut bacterial strain consortium in gnotobiotic mice. We produced 'Microbiota Functional Activity Biosensors' (MFABs) consisting of glycans covalently linked to the surface of fluorescent paramagnetic microscopic glass beads. Three MFABs, each containing a unique glycan/fluorophore combination, were simultaneously orally gavaged into gnotobiotic mice, recovered from their intestines, and analyzed to directly quantify bacterial metabolism of structurally distinct arabinans in different human diet contexts. Colocalizing pea-fiber arabinan and another polysaccharide (glucomannan) on the bead surface enhanced in vivo degradation of glucomannan. MFABs represent a potentially versatile platform for developing new prebiotics and more nutritious foods.

**\*For correspondence:**
jgordon@wustl.edu

## Introduction

Increasing effort is being directed to determining how components of diets consumed by various human populations impact the composition and expressed functional features of their gut microbial communities (e.g., *Johnson et al., 2019*; *Ghosh et al., 2020*). A hoped-for benefit is to gain new insights about how food ingredients and their biotransformation by the microbiota are linked to various aspects of human physiology, and new ways to both define and improve nutritional status. However, there are many formidable challenges. The gut microbiota is complex and dynamic and exhibits considerable intra- and interpersonal variation in its configurations (*Lloyd-Price et al., 2017*). The chemical compositions of food staples are being cataloged at ever-deepening levels of detail using higher throughput analytical methods, such as mass spectrometry. Even as this knowledge is being acquired, how food components are recognized by members of the microbiota, and the pathways through which these chemical entities are metabolized by community members to influence their functions and those of the host remain challenging to define. Furthermore, much needs to be learned about the effects of current methods of food processing on the representation of bioactive food components (*Carmody et al., 2015*; *Wolf et al., 2019*), and about the mechanisms that determine whether and how microbes compete and/or cooperate for them.

Plant fibers epitomize these challenges. Plant fibers are complex mixtures of biomolecules whose composition varies depending upon their source, method of initial recovery, and the food processing

**eLife digest** Tens of trillions of microbes living in the gut help humans and other animals digest their food. In the process, the microbes provide necessary nutrients for themselves and the animal. Learning more about the interaction of food components and gut bacteria could help scientists to better understand how different diets affect human health. Currently, studying these complex interactions is challenging, but new technologies that measure microbial nutrient processing in the gut could help.

Now, Wesener et al. show that swallowable microscopic biosensors can measure how gut bacteria break down nutrients from food. To make the biosensors, Wesener et al. attached complex carbohydrates extracted from peas and fluorescent tags to microscopic beads. In the experiments, mice colonized with human gut microbes were fed the beads along with a traditional low fiber, Western diet. Some of the animals also received fiber supplements.

The microscopic beads were then recovered from the intestines after digestion and the remaining carbohydrates on the beads were measured. The genetic makeup of the gut microbiome and the expression of microbial genes was also examined. The experiments revealed which pea carbohydrates the gut microbes consumed and showed that pairing certain carbohydrates together on the microbead surface increased their digestion in mice that received fiber supplements.

If future studies prove that the microbead biosensors created by Wesener et al. are safe for humans to ingest, they could be used to help diagnose how well a person's gut microbiota can process different foods. Studies using the microbead sensors may also help scientists develop more nutritious foods or supplements that promote the growth of microbes important for health.

techniques used to incorporate them into food products that have satisfactory organoleptic properties (texture, taste, smell) (*Shahidi and Ho, 1998*; *Caffall and Mohnen, 2009*). Plant fibers are composed of but not limited to polysaccharides, proteins, fatty acids, polyphenols, and other plant-derived small molecules (*Nicholson et al., 2012*; *Scalbert et al., 2014*). Isolating and/or purifying component polysaccharides from crude plant fiber mixtures for studies of the mechanisms by which they influence members of the microbiota can be very challenging; even if separation is achieved, painstaking analysis is required to define their structures (*Pettolino et al., 2012*). Nonetheless, knowledge of the genetic underpinnings of how various members of the gut microbiota recognize and metabolize glycans has increased substantially in recent years through a combination of in vitro and in vivo approaches (*Brown and Koropatkin, 2020*; *Kaoutari et al., 2013*; *Ndeh and Gilbert, 2018*). Knowing that a given microbiota member has a suitable complement of genes for acquiring and processing a given glycan structure in vitro does not necessarily predict whether that organism will be a consumer in vivo (*Flint et al., 2015*). For example, an individual's microbiota may harbor a number of organisms with the capacity to compete or cooperate with one another for utilization of a given type of glycan. Additionally, the physical and chemical structure of a plant fiber (e.g., its size, surface properties, nutrient composition) in a given region of the gut could influence which set of microbes attach to its surface, how its associated microbes prioritize consumption of its component glycans, and how/whether fiber particle-associated microbes can share products of glycan metabolism with one another.

The current study illustrates an approach for identifying bioactive molecules from plant fibers and defining how they affect and how they are processed by members of the human gut microbiota. Pea fiber was selected based on results obtained from a screen we conducted of 34 types of food-grade crude plant fibers obtained from various sources, including the byproducts of food manufacturing (*Patnode et al., 2019*). The screen was performed in gnotobiotic mice colonized with a defined consortium of cultured sequenced human gut bacterial strains, including several saccharolytic Bacteroides species. Mice were fed a low-fiber diet formulated to represent the upper tertile of saturated fat consumption and lower tertile of fruit and vegetable consumption by individuals living in the USA, as reported in the NHANES database (*Ridaura et al., 2013*). Supplementation of this 'HiSF-LoFV' diet with fiber generated from the seed coat of the pea, *Pisum sativum*, produced a significant increase in the abundance of *Bacteroides thetaiotaomicron* (*Patnode et al., 2019*). We have now isolated a bioactive arabinan-enriched fraction from pea fiber, defined its structure, and

characterized how a model human gut community, containing human gut Bacteroides established in gnotobiotic mice, responds to the isolated arabinan-enriched fraction versus unfractionated pea fiber. We go on to describe a generalizable method for covalently attaching different glycans to microscopic paramagnetic glass beads with different covalently bound fluorophores, eliminating the proteinaceous component of an approach published by our group that relies on streptavidin-coated beads and bifunctional biotin-conjugated polysaccharides (*Patnode et al., 2019*). Introduction of these 'Microbiota Functional Activity Biosensors' (MFABs) into gnotobiotic mice fed the HiSF-LoFV diet with or without glycan supplementation followed by their recovery from the gut allowed us to directly compare the capacity of these glycans to be degraded by this community. Chemically colocalizing pea fiber arabinan with glucomannan, another type of polysaccharide not found in the diet, in a monolayer on an MFAB surface enhanced the efficiency of microbial community degradation of bead-associated glucomannan when animals were given a pea-fiber-supplemented HiSF-LoFV diet. These findings illustrate how knowledge of the bioactive components of fibers, and the capacity to directly measure microbiota function with MFABs, could provide new approaches for designing 'next-generation' prebiotics and foods that are more accessible to, and have a greater impact on, the gut microbiota (and by extension, the host).

## Results

### Isolation of a 2-*O*-branched arabinan fraction from pea fiber

Raw pea fiber was subjected to serial extraction with aqueous buffers (*Pattathil et al., 2012*). Eight soluble fractions were assayed for total nucleic acid, protein, and carbohydrate content plus monosaccharide composition (*Supplementary file 1*). The fraction isolated under the harshest conditions (fraction 8; 4M KOH) possessed high carbohydrate and low protein content, high monosaccharide diversity, and a high relative proportion of arabinose. We subsequently developed a scalable abbreviated procedure for preparing this fraction from pea fiber (*see Materials and methods*). The mole percent representation of arabinose in the end product was 71% (corresponding values for xylose, galactose, and glucose were 12%, 11% and 6%, respectively) (*Figure 1A*). Glycosyl-linkage analysis of the isolated glycan revealed a 2-*O*-branched arabinan (5-substituted arabinose, 2,5-substituted arabinose, 2,3,5-substituted arabinose) (*Figure 1B*) attached to small rhamnogalacturonan-I (RGI) pectic fragments (2-substituted and 2,4-substituted rhamnose) and small galactan oligomers (4-substituted galactose) (*Supplementary file 1*). Xylose was present as a linear xylan polysaccharide (4-substituted xylose) and glucose in the form of residual starch. We named the isolated polysaccharide fraction 'pea fiber arabinan' (PFABN) based on (1) these linkage results, (2) the fact that arabinose comprises the majority of its monosaccharide content (71 mole percent), and (3) our observation that ~90% of all non-starch carbohydrate is represented by what is likely a single species of polysaccharide, with the remaining being xylan.

### In vivo effects of PFABN on human gut Bacteroides

The biological activity of PFABN was compared to that of unfractionated pea fiber and arabinan isolated from sugar beet. Unlike PFABN, sugar beet arabinan (SBABN) is primarily 3-*O*-branched as revealed by glycosyl-linkage analysis (*Figure 1B*, *Supplementary file 1*). PFABN also contains twofold more triply branched 2,3,5-substituted arabinose monomers, suggesting it is more sterically encumbered than SBABN. The galactan portion of each arabinan is also unique: while both polymers contain 4-substituted galactose, 4,6-substituted galactose is enriched more than twofold in PFABN, while 6- and 3,6-substituted galactose are enriched threefold and fourfold in SBABN, respectively (*Supplementary file 1*). In vitro assays performed in a minimal defined medium (*McNulty et al., 2013*) with Bacteroides type strains established that *B. ovatus* ATCC 8483, *B. cellulosilyticus* WH2, and *B. thetaiotaomicron* VPI-5482 grew on isolated PFABN and SBABN, although less rapidly during the exponential phase of growth and to a lower cell density than in medium containing an equivalent concentration of D-glucose (results based on measurements of OD600; see *Figure 1C*, *Supplementary file 1*). In contrast, *B. vulgatus* ATCC 8482 grew as rapidly or faster and to the same or higher density on PFABN and SBABN, respectively, compared to D-glucose ($p < 0.05$, one-way analysis of variance [ANOVA] with Tukey's honest significant difference, FDR corrected). *B. vulgatus*

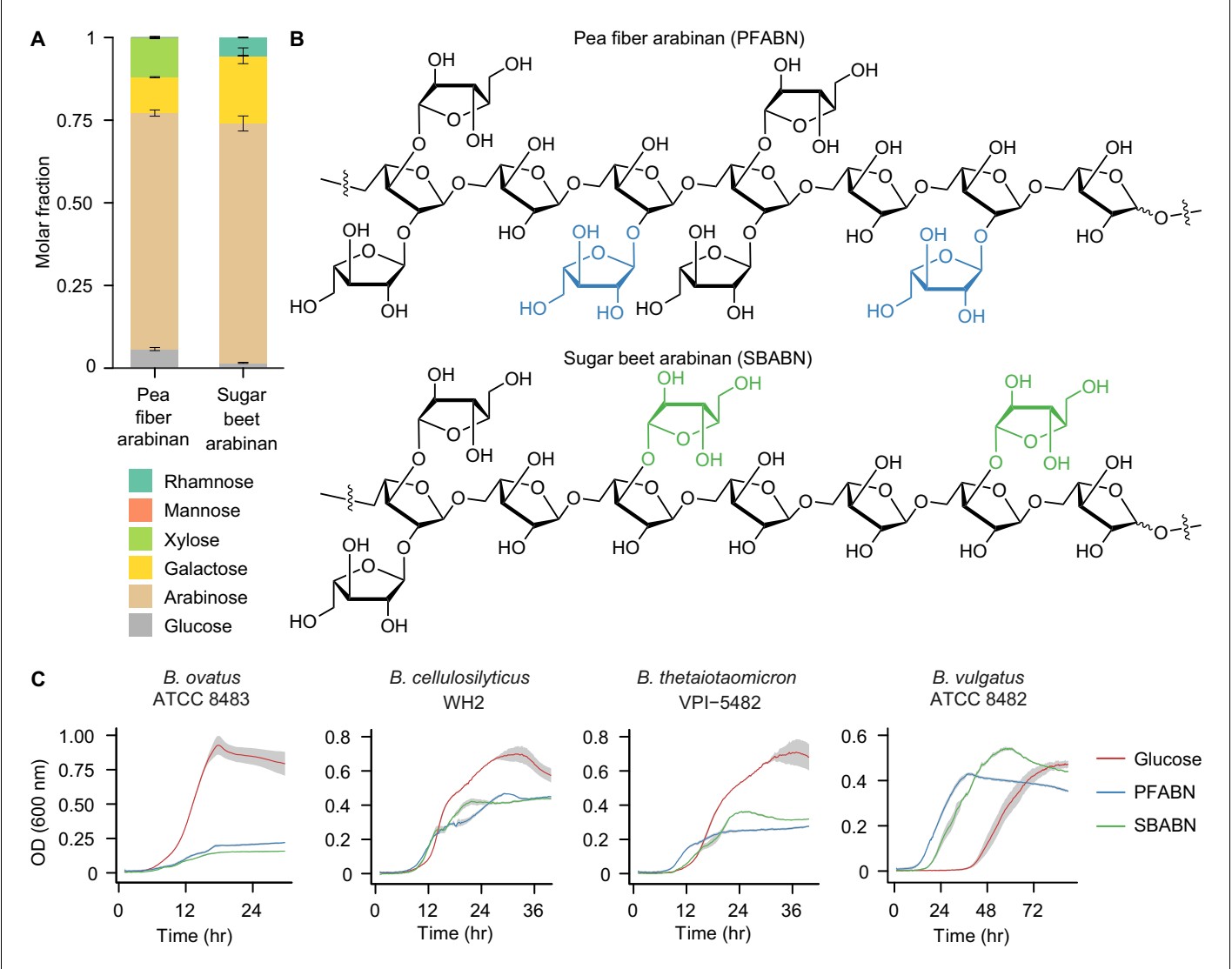

**Figure 1.** Isolation of an arabinose-rich fraction from pea fiber. (A) Mole fraction of neutral monosaccharides from PFABN and SBABN. Mean values ± s.d. from triplicate technical measurements are shown. (B) Proposed structure of arabinan isolated from pea fiber and sugar beet as determined by glycosyl-linkage analysis. The colored arabinose monosaccharides highlight the different glycosyl branching patterns found in arabinan isolated from the two sources. (C) Growth curves of four *Bacteroides* strains cultured in minimal medium containing either glucose or arabinan as the sole carbon source. Data from samples where no exogenous carbon source was added are subtracted from all curves. Solid lines represent the mean and shaded area the s.e.m. of quadruplicate cultures (data shown are representative of 3 independent experiments). (See also ***Supplementary file 1***).

ATCC 8482 also reached exponential growth more quickly than when D-glucose was present in this medium (***Figure 1C***).

Adult germ-free C57Bl/6J mice were colonized with a 14-member consortium of sequenced bacterial strains containing 58,537 known or predicted protein-coding genes. The consortium included the four Bacteroides type strains, another strain of *B. thetaiotaomicron*, three additional Bacteroides species (*B. caccae*, *B. massiliensis*, *B. finegoldii*) plus six other types of bacteria (***Figure 2A***). Five of the Bacteroides (*B. thetaiotaomicron* strains VPI-5482 and 7330, *B. vulgatus* ATCC 8482, *B. cellulosilyticus* WH2, *B. ovatus* ATCC 8483) were each represented by previously described libraries of tens of thousands of transposon (Tn) insertion mutants (***Hibberd et al., 2017***; ***Wu et al., 2015***). These studies had shown that collectively the overall change in abundance of each mutant library in response to various diet manipulations was similar to the corresponding parental wild-type strain.

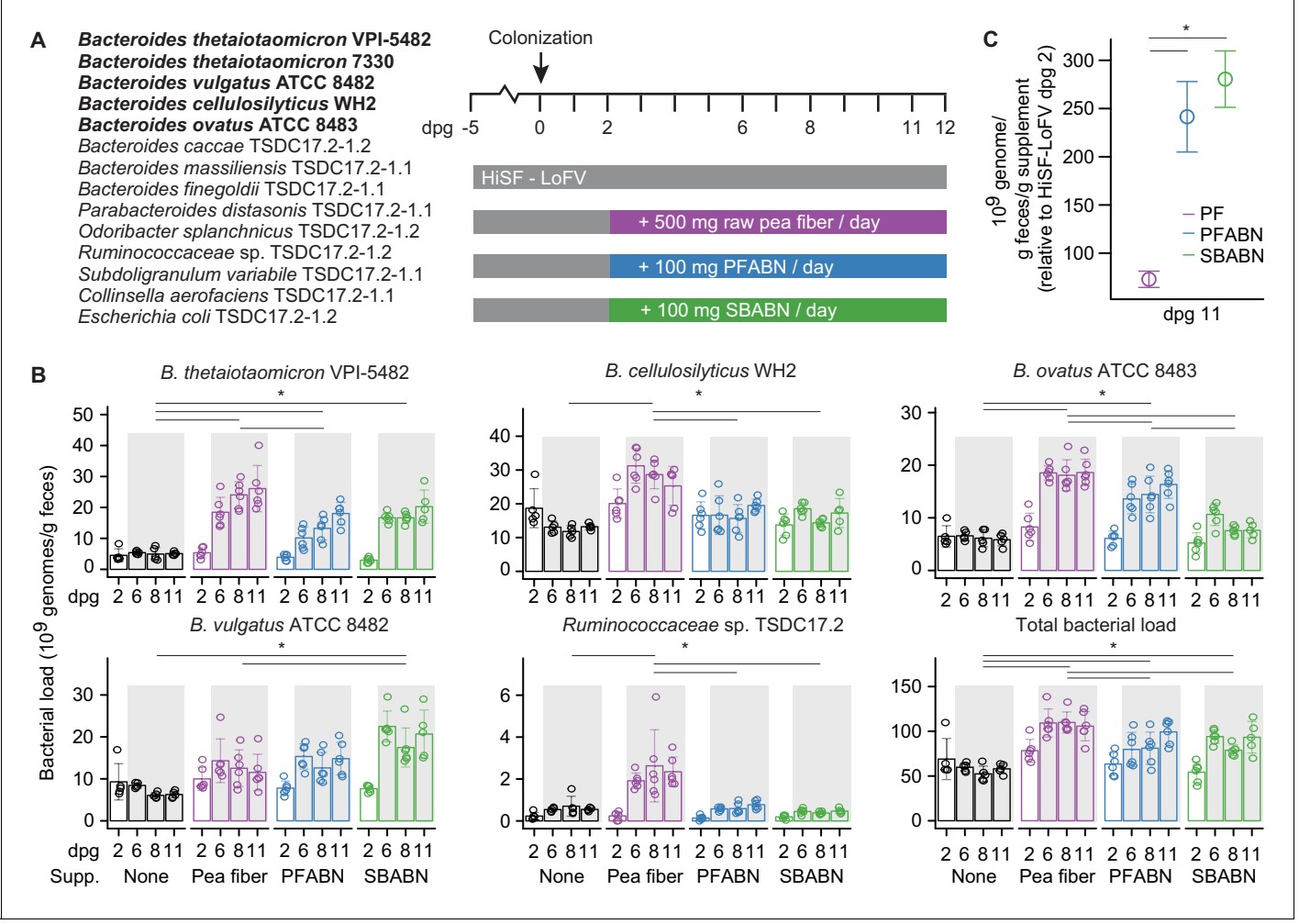

**Figure 2.** Assessing the biological activity of PFABN in gnotobiotic mice colonized with a defined consortium of human gut bacterial strains. (**A**) Experimental design. Germ-free mice were fed the unsupplemented HiSF-LoFV diet for 5 days then colonized with the indicated group of 14 bacterial strains (the five Bacteroides strains represented in the form of Tn mutant libraries are highlighted in boldface). Two days after gavage of the consortium different groups of animals were switched to a HiSF-LoFV diet supplemented with raw pea fiber, PFABN or SBABN and fed that diet monotonously for 10 days (average dose of supplement consumed/per day is shown). Control animals were maintained on the unsupplemented HiSF-LoFV diet. (**B**) Absolute abundances of supplement-responsive bacterial strains, plus the total bacterial load of all 14 strains in fecal samples obtained at the indicated time points (each dot represents a single animal; bar height represents the mean; error bars represent s.d.). *p<0.01 for comparisons denoted by horizontal lines (generalized linear mixed-effects model [Gaussian]; two-way ANOVA with Tukey's HSD, FDR corrected; the data shown are from Experiment two in *Supplementary file 2* and are representative of two independent biological experiments). (**C**) Specific activity of each diet supplement on the summed total absolute abundances of the four diet-responsive Bacteroides. Open circles represent mean values and error bars the s.e.m. of two independent biological experiments (n = 10–11 mice/treatment arm). *p<0.01 for comparisons defined by the horizontal lines (generalized linear mixed-effects model [Gaussian]; two-way ANOVA with Tukey's HSD, FDR corrected). (See also *Supplementary files 2–4*).

The online version of this article includes the following source data and figure supplement(s) for figure 2:

**Source data 1.** Identifying PULs that function as key fitness determinants in the different diet contexts.

**Figure supplement 1.** The effects of supplementing the HiSF-LoFV diet with unfractionated pea fiber, PFABN, or SBABN on PUL gene expression.

Two days post-gavage (dpg 2) of the defined bacterial consortium, mice were switched from the unsupplemented HiSF-LoFV diet to a HiSF-LoFV diet supplemented with either unfractionated pea fiber, isolated PFABN, or isolated SBABN. Animals were fed these diets ad libitum for 10 days and then euthanized. A control group was maintained on unsupplemented HiSF-LoFV chow (*Figure 2A*; n = 5–6 mice/treatment group). The amount of supplementation was calibrated so that mice in all treatment arms would receive the same daily dose of arabinan (100 mg; see Materials and methods).

The absolute abundances of each strain were defined by shotgun sequencing of DNA isolated from feces collected on dpg 2, 6, 8, and 11 and from the cecum at the time of euthanasia (two independent experiments, labeled Experiment one and Experiment two in *Supplementary file 2*; *McNulty et al., 2011*; *Stämmler et al., 2016*). Of the 14-member consortium, only *B. thetaiotaomicron* 7330 failed to colonize mice at levels above a (relative) abundance of 0.1% under any of the diet conditions tested.

Five community members exhibited statistically significant, albeit distinct, differences in their absolute abundances in response to specific glycan preparations: (1) *B. thetaiotaomicron* VPI-5482 responded to all three supplements; (2) *B. ovatus* ATCC 8483 increased after exposure to both intact pea fiber and PFABN but not to SBABN; (3) *B. vulgatus* ATCC 8482 exhibited significant increases with SBABN; and (4) *B. cellulosilyticus* WH2 and *Ruminococcaceae sp.* TSDC 17.2 increased significantly when exposed to intact pea fiber but not to either of the isolated arabinan preparations (*Figure 2B*, *Supplementary file 2*) ($p < 0.01$, generalized linear mixed-effects model [Gaussian]; two-way ANOVA with Tukey's HSD, FDR corrected). We calculated the sum total of the changes in absolute abundances of the four responsive Bacteroides relative to dpg two under each diet condition and expressed the results as bacterial cells per gram of feces per gram of supplement provided to each mouse daily. This metric of specific activity, which takes into account the absolute mass of supplement consumed daily by members of the different groups of mice, revealed that both of the isolated arabinan preparations had a significantly greater effect compared to intact pea fiber at the doses tested (*Figure 2C*) ($p < 0.05$ mixed-effects linear model [Gaussian]; one-way ANOVA with Tukey's honest significant difference, FDR corrected).

We took advantage of the fact that the gene content of the community was known and performed mass spectrometry-based metaproteomic analysis on feces, collected on dpg 6, to define the responses of community members to the different glycan preparations. Bacteroides spp. possess multiple polysaccharide utilization loci (PULs); a shared feature of PULs is an adjacent pair of *susC* and *susD* homologs responsible for binding extracellular polysaccharide fragments and importing them into the periplasm. PUL genes also encode various carbohydrate active enzymes (CAZymes) involved in polysaccharide depolymerization, as well as transcriptional regulators that allow the locus to be induced in the presence of glycans it can recognize/utilize (*Anderson and Salyers, 1989*; *Terrapon et al., 2018*). Regulated expression of PULs allows bacteria to acquire nutrients within the highly competitive environment of the gut (*Martens et al., 2011*; *Tuncil et al., 2017*).

Supplemental results, *Figure 2—figure supplement 1*, *Figure 2—source data 1*, and *Supplementary file 3* summarize the results of our analysis of PUL protein expression from the two independent experiments (total of 10–11 mice/treatment arm). Based on gene set enrichment analysis (*Luo et al., 2009*), we identified 14, 12, 11, and 8 PULs that we deemed 'responsive' to at least one of the diet supplements in *B. thetaiotaomicron* VPI-5482 (BT), *B. ovatus* ATCC 8483 (Bovatus), *B. cellulosilyticus* WH2 (BcellWH2), and *B. vulgatus* ATCC 8482 (BVU), respectively (adjusted p-value<0.05, unpaired one-sample Z-test, FDR-corrected).

Additionally, we used multi-taxon insertion site sequencing (INSeq, *Wu et al., 2015*) of the five strains represented as Tn mutant libraries to identify genes with significant contributions to bacterial fitness in each diet context. Fitness was calculated as (1) the log2 ratio of the number of sequencing reads originating from the site of insertion of the Tn in the organism in fecal communities sampled on dpg 6 versus dpg 2, relative to (2) the same ratio calculated in mice monotonously fed the unsupplemented HiSF-LoFV diet. A negative score indicates that a gene is important for fitness. The score of each gene was parameterized using linear models generated with limma (*Ritchie et al., 2015*) to identify those whose effects on fitness were significantly different compared to when the unsupplemented HiSF-LoFV diet was being consumed. The results disclosed that the fitness scores of a total of 39 genes in *B. thetaiotaomicron* VPI-5482, 135 genes in *B. ovatus* ATCC 8483, 346 genes in *B. cellulosilyticus* WH2, and 82 genes in *B. vulgatus* ATCC 8482 were significantly decreased during diet supplementation with either pea fiber, PFABN, or SBABN (*Supplementary file 4*) (adjusted p-value<0.05, FDR corrected). Plots of fitness score versus change in protein abundance were generated for all genes in each of these Bacteroides. Supplemental results and parts A–D of *Figure 2— source data 1* summarize how expression and the fitness contribution of specific PULs vary for individual Bacteroides across the dietary contexts tested. Together, these community configurational and functional responses to diet supplementation provided evidence that PFABN is a key bioactive component of pea fiber utilized by *B. thetaiotaomicron* VPI-5482, *B. vulgatus* ATCC 8482, *B.*

*cellulosilyticus* WH2, and *B. ovatus* ATCC 8483. However, these results do not directly establish that it is consumed, nor do they offer a direct comparison of the efficiency of metabolism of PFABN and SBABN. To produce such evidence, we developed a bead-based method for quantifying polysaccharide degradation within the intestinal tracts of colonized gnotobiotic mice.

## Microbiota functional activity biosensors

### Covalent linkage of various fluorescent labels and glycans to paramagnetic MFABs

To quantify PFABN and SBABN degradation as a function of diet, we sought a versatile way to covalently link polysaccharides to recoverable, paramagnetic, microscopic glass beads that could function as biosensors of their degradation. For covalent polysaccharide immobilization on a bead surface, we leveraged a cyano-transfer reaction employed in the synthesis of polysaccharide-conjugate vaccines (*Lees et al., 1996*; *Shafer et al., 2000*).

*Figure 3A, B* outlines the procedure for generating fluorescently labeled, polysaccharide-coated beads. *First*, the surfaces of 10 μm diameter glass beads were sialyated by reaction with an amine- and/or phosphonate-organosilane (Step one in *Figure 3A*). This approach provided us with control over the stoichiometry and properties of surface functional groups (amine and phosphonate) to be used for further derivatization with a fluorophore and ligand immobilization. We found that coating with a 1:1 mole ratio of (3-aminopropyl)triethoxysilane (APTS) and 3-(trihydroxysilyl)propyl methylphosphonate (THPMP) to install both amine and phosphonate functional groups on the bead surface provided a nucleophilic handle and decreased nonspecific ligand binding and bead aggregation (*Bagwe et al., 2006*). Surface sialyation and the amount of reactive surface amine functional groups were monitored by measuring the zeta potential of beads following organosilane derivatization with or without amine acetylation (*Figure 3—figure supplement 1A*). Surface amine functional groups were subsequently quantified using a ninhydin-based colorimetric assay (*Soto-Cantu et al., 2012*). The results revealed that amine plus phosphonate functionalized beads contain $2.18 \times 10^{10} \pm 3.49 \times 10^9$ (mean ± s.d.) reactive amines per bead versus $1.20 \times 10^9 \pm 1.92 \times 10^8$ reactive amines after surface acetylation (*Figure 3—figure supplement 1B*). By comparison, streptavidin-coated beads used in our previously published procedure (*Patnode et al., 2019*) possess 23-fold fewer potential binding sites ($9.21 \times 10^8 \pm 1.13 \times 10^8$ molecules of biotin per bead). *Second*, we attached unique fluorogenic tags directly to the bead surface so that multiple bead types with different immobilized polysaccharides could be analyzed simultaneously within a given gnotobiotic animal. To do so, surface-modified beads were reacted with an *N*-hydroxysuccinimide (NHS) ester-activated fluorophore (Step two in *Figure 3A*). Fluorophore coupling was specific to beads with surface amines (*Figure 3—figure supplement 1C*). Bead fluorescence could be modulated over four orders of magnitude simply by titration of the reactant fluorophore (*Figure 3—figure supplement 1D*). Low levels of fluorophore immobilization on beads not coated with APTS or on acetylated beads likely reflect incomplete acetylation with acetic anhydride or nonspecific fluorophore adsorption. *Third*, polysaccharide was activated by reaction with 1-cyano-4-dimethylaminopyridinium tetrafluoroborate (CDAP) to generate an electrophilic cyanate-ester intermediate (*Figure 3B*); activated polysaccharide reacts with amines on the surface of the amine plus phosphonate bead. *Lastly*, a hydride reduction was performed to reduce any Schiff base formed with the polysaccharide reducing end and likely the resultant isourea bond (see *Figure 3—figure supplement 2* for a schematic of the surface of an arabinan-coated MFAB).

Using SBABN as a test case, we found that adding 0.2 mg of CDAP per mg polysaccharide resulted in consistent and specific SBABN immobilization without ligand overactivation (manifested by aggregation and carbamoylation of hydroxyl groups) (*Figure 3C*). Immobilization was dependent upon the presence of reactive surface amines; amine acetylation with acetic anhydride reduced surface amine functional groups, fluorophore conjugation, and polysaccharide immobilization (*Figure 3C*, *Figure 3—figure supplement 1B–D*). Therefore, acetylated beads were used as controls in the studies described below. Throughout our experiments, levels of conjugation ranged from 2 to 20 ng of arabinose per 1000 beads when immobilizing SBABN. Conjugation proceeded as expected based on the p$K_a$ of the bead amine: pH 7.5–7.8 yielded maximal immobilization (high pH results in cyanate-ester hydrolysis, while low pH favors amine protonation) (*Figure 3—figure supplement 3A*). Conjugation efficiency was not significantly different when different buffer solutes were tested

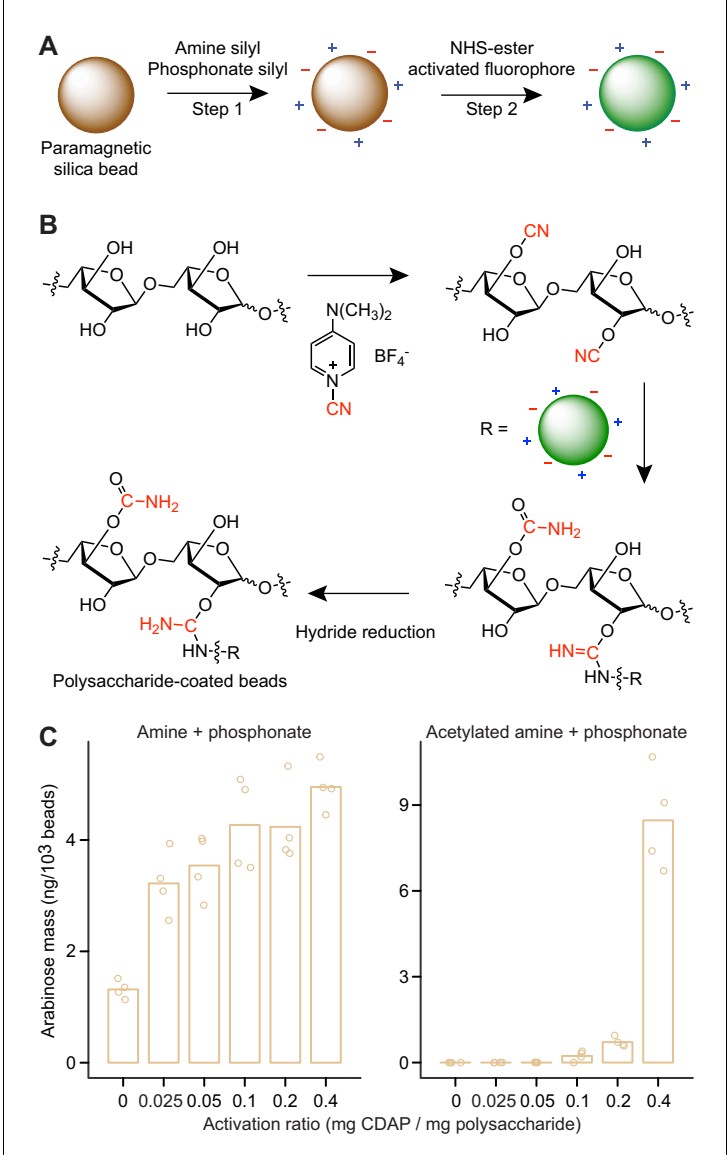

**Figure 3.** Generating microscopic paramagnetic glass beads with covalently attached fluorophores and glycans. (A,B) Steps used for producing MFABs. The transferred cyano-group from 1-cyano-4-dimethylaminopyridinium tetrafluoroborate (CDAP), and its modification during ligand immobilization are highlighted in red (panel b). Arabinose oligosaccharide is shown as a representative ligand for immobilization. Amine and phosphonate functional groups are denoted by '+' and '–' symbols, respectively. See *Figure 3—figure supplement 2* for a more complete depiction of the chemical linkages represented on the surface of an MFAB with bound arabinan. (C) Arabinose released during acid hydrolysis from amine plus phosphonate beads with and without surface amine groups acetylated. Beads were coated with SBABN that had been activated using increasing mass ratios of CDAP. Each point represents a single technical measurement (n = 4). Bar height represents the mean value.

The online version of this article includes the following figure supplement(s) for figure 3:

**Figure supplement 1.** Characterizing the modified surface chemistry of paramagnetic glass beads.
**Figure supplement 2.** Schematic of a fluorescent arabinan-coated MFAB.
**Figure supplement 3.** Conjugation reaction conditions influence immobilization of polysaccharides on the surfaces of the paramagnetic glass beads.

(*Figure 3—figure supplement 3B*). A low and inconsistent level of polysaccharide could be conjugated onto a bead surface in the absence of CDAP (*Figure 3C*), likely through reductive amination of the polysaccharide reducing end. Lastly, using maltodextrin as a model, we tested whether oligosaccharides could also be immobilized using CDAP activation and capture onto amine plus phosphonate beads. Maltodextrin (dextrose equivalent 13–17; estimated $M_n$ ~1300 Da) was activated with CDAP (0.2 mg/mg oligosaccharide) prior to attachment to the bead surface. Conjugation resulted in $4.43 \pm 1.03$ ng of immobilized glucose per 1000 beads or $2.05 \times 10^9$ molecules of maltodextrin per bead (*Figure 3—figure supplement 3C*); this represents 9.4% of all available surface reactive amines.

## Quantifying polysaccharide degradation with MFABs in vitro

To determine whether CDAP-immobilized polysaccharides are good proxies for hydrated plant cell wall fragments and polysaccharide components (i.e. are anchored polysaccharides accessible to bacterial glycosidases that generate oligosaccharide inducers of PULs), we first performed several in vitro experiments. We initially incubated PFABN-coated beads with glycosyl hydrolyses and the fraction of arabinan remaining on the bead surface was measured by gas chromatography–mass spectrometry (GC–MS). Incubation with a combination of endo-arabinanase and α-L-arabinofuranosidase enzymes removed $88 \pm 1.2\%$ (mean $\pm$ s.d.) of the immobilized PFABN after 30 min while incubation with each enzyme individually removed greater than 80% of the glycan after 30 min and more than 90% after 20 hr. In contrast, an endo-inulinase with specificity toward β(1-2)-linked fructose resides failed to degrade PFABN (*Figure 4—figure supplement 1*). These results indicate that a substantial portion of the bead-immobilized glycan was susceptible to enzyme-catalyzed degradation.

As a prelude to in vivo studies that would test the ability of MFABs to measure the saccharolytic activity of the defined community in defined dietary contexts, we next incubated PFABN-coated MFABs with *B. thetaiotaomicron* VPI-5482 or *B. cellulosilyticus* WH2 grown to mid-log phase in defined minimal medium (*McNulty et al., 2013*). Saccharolytic activity was measured by using GC–MS to quantify the mass of arabinose retained on beads as a function of incubation time and whether or not soluble PFABN was included in the culture medium (reference control: beads recovered from incubations lacking bacteria). Under the conditions tested, added PFABN was required for degradation of immobilized PFABN from beads; the amount of PFABN removed from the bead surface increased with increasing incubation time, with maximum values of $64 \pm 9\%$ (mean $\pm$ s.d.) and $48 \pm 3\%$ achieved with *B. thetaiotaomicron* VPI-5482 and *B. cellulosilyticus* WH2, respectively (*Figure 4—figure supplement 2*). The effect of supplemented PFABN in this in vitro MFAB-based degradation assay is consistent with our in vivo metaproteomic analysis, demonstrating its ability to induce PULs involved in its utilization (*Figure 2—figure supplement 1*). Together, these results provided further evidence of the accessibility of bead-bound polysaccharide to degradation by secreted or bacterial cell surface-associated glycoside hydrolases.

## Quantifying polysaccharide degradation with MFABs in gnotobiotic mice

PFABN and SBABN were immobilized onto amine plus phosphonate-derivatized beads. Beads acetylated with acetic anhydride after fluorophore labeling were used as controls (*Figure 4A*). Each of these three bead types contained a unique fluorophore. The three bead types were pooled, and the mixture was introduced by oral gavage into four groups of mice 10 days after they received the 14-member consortium: one group of recipient animals had been fed the unsupplemented HiSF-LoFV diet, while the other groups had received HiSF-LoFV containing unfractionated pea fiber, PFABN, or SBABN (n = 5 animals/group). Germ-free mice fed HiSF-LoFV supplemented with PFABN served as controls (n = 5; Experiment 1). The bead mixtures were harvested using a magnet from the cecums of animals four hours after their introduction by oral gavage; the individual bead types were then purified by fluorescence-activated cell sorting (FACS). Polysaccharide degradation was quantified by GC–MS of neutral monosaccharides released after acid hydrolysis of the purified beads. Results were referenced to the masses of monosaccharides released from aliquots of each input bead type (i.e., the same bead preparation but never introduced into mice).

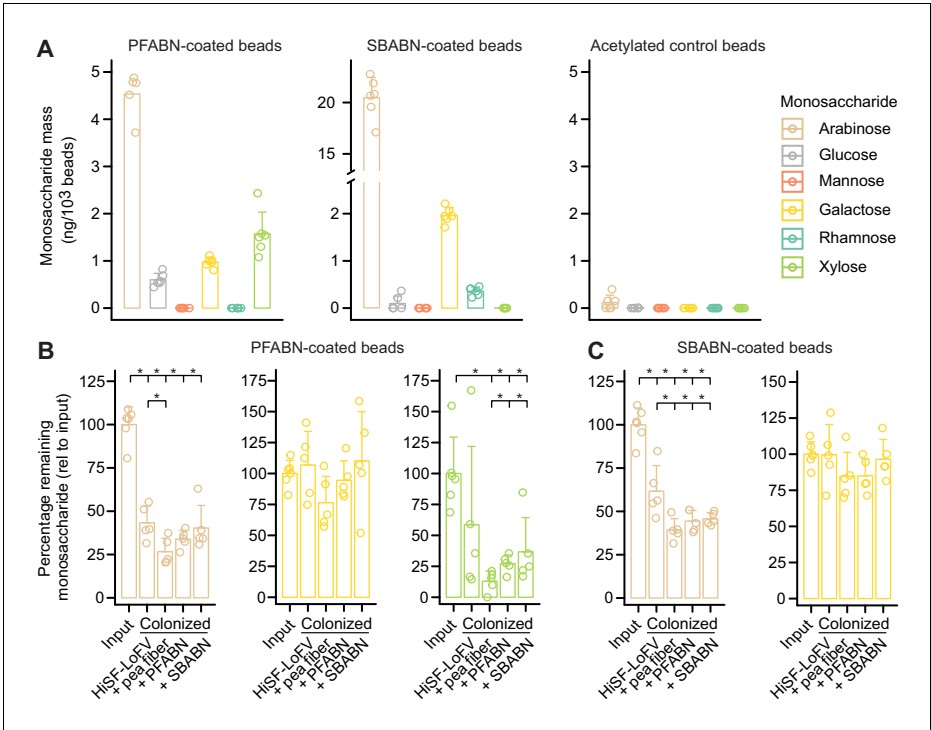

**Figure 4.** Quantifying microbial degradation of PFABN- and SBABN-coated beads in colonized gnotobiotic mice fed unsupplemented or supplemented HiSF-LoFV diets. (**A**) Monosaccharide composition of beads containing covalently bound PFABN or SBABN. Control beads were subjected to surface amine acetylation. The amount of monosaccharide released after acid hydrolysis was quantified by GC-MS. Each point represents a single measurement. Bar height denotes the mean while error bars represent the s.d. (n = 6 biological replicates). (**B,C**) Percentage of arabinose, galactose and xylose remaining on the surface of beads recovered from the cecums of mice fed the indicated diets (n = 5 mice/treatment group). Each point represents a single animal. Bar height denotes the mean while error bars represent the s.d. p<0.05 (Mann–Whitney U test compared to the group furthest to the left). (See also *Supplementary file 5*).

The online version of this article includes the following figure supplement(s) for figure 4:

**Figure supplement 1.** Enzyme degradation of PFABN immobilized on an MFAB surface using CDAP chemistry.

**Figure supplement 2.** Degradation of MFAB-bound PFABN by *B. thetaiotaomicron* VPI-5482 and *B. cellulosilyticus* WH2 in vitro.

**Figure supplement 3.** Assaying whether bead-linked polysaccharides are degraded in germ-free mice.

The quantities of neutral monosaccharides liberated by acid hydrolysis from the surfaces of beads recovered from the cecums of germ-free mice were not significantly different from the amounts liberated from the input bead preparations with one exception – a slight, albeit statistically significant, increase in galactose from beads coated with PFABN and SBABN (*Figure 4—figure supplement 3*; p<0.05, Mann–Whitney U test). This result established the stability and utility of cyanate-ester–coupled MFABs for studying polysaccharide degradation within the mouse gut, and the recalcitrance of both PFABN and SBABN to host digestive enzymes.

In contrast to germ-free controls (see *Figure 4—figure supplement 3*), the mass of arabinose was significantly decreased when PFABN- or SBABN-coated beads were recovered from colonized mice fed the unsupplemented HiSF-LoFV diet (*Figure 4B,C*; p<0.05, Mann–Whitney U test). Compared to the base HiSF-LoFV diet, supplementation with unfractionated pea fiber induced a community configuration associated with significantly increased capacity to degrade both PFABN and SBABN as judged by the amount of arabinose remaining on recovered beads (*Figure 4B,C*; p<0.05, Mann–Whitney U test). Loss of arabinose from either PFABN- or SBABN-beads was not significantly different between the two bead types when the HiSF-LoFV diet was supplemented with either of these isolated arabinan preparations, demonstrating functional equivalence in the capacity of each community to utilize either arabinan (see *Figure 4B,C* and *Supplementary file 5* which provides

evidence that results from cecal samples [Experiment 1] and fecal samples [Experiment 2] were comparable).

Beads coated in PFABN revealed that xylan (xylose monosaccharide remaining on PFABN beads) was more efficiently processed by the microbiota in all three supplemented diet contexts (*Figure 4B*; p<0.05, Mann–Whitney U test). Our group previously explored the importance of arabinoxylan utilization from the base HiSF-LoFV diet (*Patnode et al., 2019*). In contrast to xylan, the galactan content from both arabinan preparations was not utilized under any of the diet conditions tested (*Figure 4B,C* and *Supplementary file 5*), suggesting that β(1-4) galactan degradation in vivo has lower priority compared to the available arabinan (*Tuncil et al., 2017*).

## Colocalization of distinct glycans on the same bead

As noted in Introduction, plant fibers have complex physical–chemical properties manifest in part by their mixtures of different glycan structures and by their varying shapes and surface properties. Plant fiber particles are impacted by methods, such as extrusion, that are commonly used to incorporate plant fibers into food products so that these products have acceptable organoleptic properties (*Caffall and Mohnen, 2009*; *Gualberto et al., 1997*), and by the mechanical forces and digestive enzymes (both host and microbial) that are encountered as food passes through the gastrointestinal tract. In foods, plant fibers exist mostly as micro-particles. Although it would be desirable to be able compare the dynamics of degradation of glycans on MFABs with glycans in food particles with similar dimensions/characteristics, robust methods for reproducibly recovering food particles from luminal contents are not currently available. Therefore, we reasoned that the MFAB platform could provide a way of testing whether deliberately colocalizing distinct polysaccharides, akin to natural plant fiber particles, would result in 'synergistic' polysaccharide degradation by microbial community members.

To explore this notion, we turned to glucomannan, a hemicellulosic linear β(1-4) polysaccharide composed of D-mannose and D-glucose. We found that among the pea fiber-responsive Bacteroides identified above, only *B. ovatus* ATCC 8483 and *B. cellulosilyticus* WH2 were able to grow in minimal medium containing glucomannan as the sole carbon source (*Figure 5A*). Both organisms have PULs known to be induced by glucomannan in vitro (PUL28 in *B. cellulosilyticus* WH2; PULs 52 and 80 in *B. ovatus* ATCC 8483); each of these PULs encodes at least one GH26 enzyme with β-mannanase activity (*Bågenholm et al., 2017*; *Martens et al., 2011*). Multiple genes in the glucomannan-responsive PUL28 of *B. cellulosilyticus* WH2 were consistently expressed, but not at significantly different levels, when mice were fed the unsupplemented HiSF-LoFV and pea fiber supplemented HiSF-LoFV diets (*Supplementary file 3*). Only two *B. ovatus* ATCC 8483 genes from its glucomannan-responsive Bovatus_PUL52 were expressed, albeit at the very limit of detection, under both diet conditions, and none from Bovatus_PUL80 (*Supplementary file 3*). Neither *B. thetaiotaomicron* VPI-5482 nor *B. vulgatus* ATCC 8482, which fail to grow on glucomannan as the sole carbon source, contain GH26, GH2, or GH130 genes with known or predicted β-mannanase or β-mannosidase activities that were induced during pea fiber supplementation (*Supplementary file 3*) (among the two organisms, only the protein products of *B. thetaiotaomicron* BT_0458 [GH2] and BT_1033 [GH130] were detected under either diet conditions, and only at the very threshold of detection).

Based on these considerations, we hypothesized that supplementing the diet with pea fiber would induce expression of PULs in community members, so that they could readily utilize bead-associated PFABN; moreover, those community members that could utilize PFABN and express β-mannanases would be able to more efficiently access/metabolize glucomannan positioned on the same bead. To test this hypothesis, we synthesized beads coated with PFABN alone, glucomannan alone, or both glycans together, as well as control acetylated beads that lack a bound polysaccharide (*Figure 5B*). These four bead types, each labeled with a distinct fluorophore, were simultaneously introduced into two groups of mice colonized with the 14-member community – one group was fed the unsupplemented HiSF-LoFV diet, while the other group received a pea fiber-supplemented diet (n = 7–8 mice/group) (*Supplementary file 2*). Beads were recovered from their cecums 4 hr after gavage; the different bead-types were then isolated using FACS (*Figure 5C*) and subjected to acid hydrolysis and neutral monosaccharide analysis by GC–MS. We used the amount of mannose remaining on the bead as a proxy of glucomannan degradation because it represents the bulk of monosaccharide present in glucomannan and is absent in PFABN. The results revealed that glucomannan on beads coated with glucomannan alone was degraded to a similar extent in mice

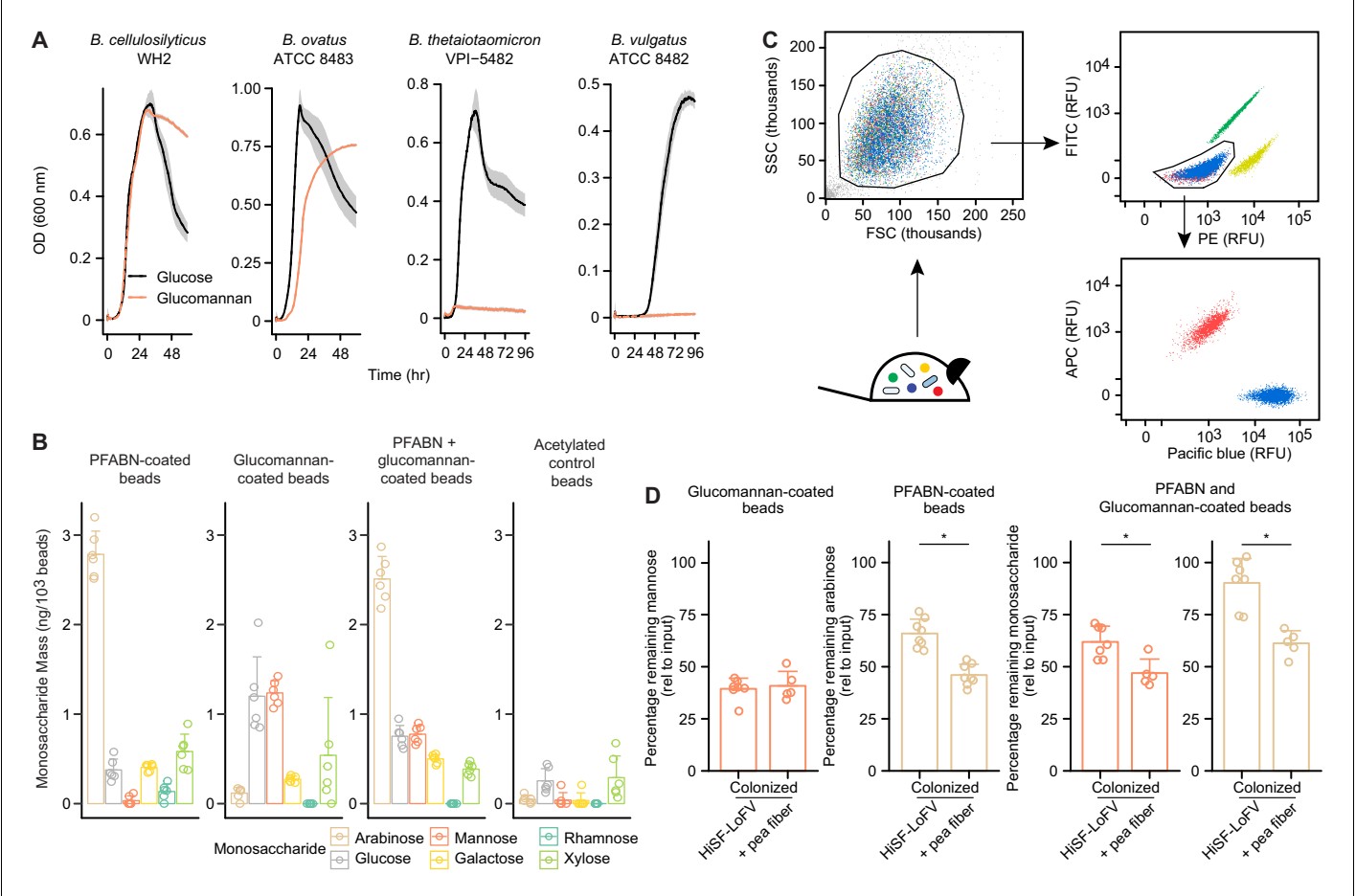

**Figure 5.** Colocalization of PFABN and glucomannan on the same bead results in augmented degradation of glucomannan in gnotobiotic mice colonized with the defined consortium and fed the pea fiber supplemented HiSF-LoFV diet. (**A**) In vitro growth of supplement-responsive Bacteroides species in minimal medium containing glucose or glucomannan as the sole carbon source. Data from samples where no exogenous carbon source was added are subtracted from all curves. The line represents the mean and shaded regions the s.e.m. of quadruplicate measurements. (**B**) Monosaccharide compositions of beads with covalently bound PFABN, glucomannan, or both PFABN and glucomannan. Control beads were subjected to surface amine acetylation. The amount of monosaccharide released after acid hydrolysis was quantified by GC-MS. Each point represents a single measurement. Bar height represents the mean and error bars the s.d. (n = 6 biological replicates). (**C**) Beads containing PFABN alone, glucomannan alone, or both glycans, as well as 'empty' acetylated control beads, each containing a unique fluorophore, were simultaneously introduced by oral gavage into gnotobiotic mice, recovered 4 hr later from their cecums. Each bead-type is subsequently purified by FACS. A representative flow cytometry plot of beads isolated from the cecum is shown. (**D**) Monosaccharide remaining on beads coated with PFABN alone, glucomannan alone, or both glycans after collection and purification from the cecums of mice fed the unsupplemented or pea fiber-supplemented HiSF-LoFV diet. Colors are identical to those used in panel b. The amount of remaining monosaccharide is expressed relative to the absolute mass of monosaccharide immobilized on the surface of each type of input bead. Each point represents a single animal. Bar height represents the mean and error bars the s.d. (n = 5–8 biological replicates). *p<0.05 (Mann–Whitney U test). (See also *Supplementary file 5*).

receiving the unsupplemented or pea fiber-supplemented HiSF-LoFV diets (*Figure 5D* and *Supplementary file 5*; p=0.87, Mann–Whitney U test). However, when presented with PFABN on the same bead, significantly more glucomannan was degraded by the microbiota of mice receiving the pea fiber-supplemented diet as compared to the unsupplemented diet (*Figure 5D*; p<0.05, Mann–Whitney U test). The amount of arabinose remaining on beads coated with PFABN and glucomannan, and PFABN alone, was also significantly reduced (degradation increased) with pea fiber supplementation (*Figure 5D*; p<0.05, Mann–Whitney U test). These results show that deliberate physical colocalization can result in quantitatively modest, albeit statistically significant synergistic degradation of polysaccharides during fiber supplementation (*Supplementary file 5*; p<0.05, linear model; diet supplement by bead-type interaction term). This finding, and the approach used to obtain these results, have implications for food science and prebiotic/synbiotic discovery efforts.

## Discussion

We have used gnotobiotic mice colonized with a defined model human gut microbial community containing 58,537 known or predicted protein-coding genes together with metaproteomic and forward genetic analyses to demonstrate the sensitivity of a microbiota to structural differences in glycans (arabinans) isolated from two distinct plant sources. Robust isolation procedures and analytical characterization are required to isolate grams of polysaccharide and test their biological effects in gnotobiotic mice. The approach we employed allowed us to isolate 50 g of material to ~85% purity with the remaining material comprised of starch and xylan. Starch can be degraded by host enzymes. The xylan present in our PFABN preparation could, in principle, influence the microbiota response. However, there is arabinoxylan already present in the base HiSF-LoFV diet, and we found that expression of known xylan-responsive PULs in *B. cellulosilyticus* WH2 and *B. ovatus* ATCC 8483 was not induced by PFABN supplementation (*Figure 2—figure supplement 1*; *Patnode et al., 2019*; *McNulty et al., 2013*; *Martens et al., 2011*). The specific activity of each isolated arabinan (in this case the increase in absolute abundance of the responsive Bacteroides per unit mass of diet ingredient consumed per day) was superior to unfractionated pea fiber (*Figure 2C*), emphasizing, as others have, the importance of characterizing the effects of plant fibers on both the microbiota and host in order to inform future studies at the intersection of food science, nutrition, and human microbial ecology research (*Gidley and Yakubov, 2019*). Although each of the isolated arabinans had distinct effects on gene expression and organismal fitness, our development of orally administered recoverable and chemically modifiable probes to quantitate community metabolic function revealed their equivalent degradation in specified diet contexts. These findings illustrate how in vivo functional assays can complement and expand current approaches for characterizing microbiomes.

The bead-based MFABs described in this report represent a technology for measuring biochemical activities expressed by a microbial community of interest in vivo or ex vivo. Future studies where MFABs are used in conjunction with genetic tools that enable rapid deletion of genes or entire multi-kilobase loci could provide new insights about the mechanisms by which community members acquire/degrade nutrients (polysaccharides). We emphasize that in this study, we use MFABs to quantify community degradative activity and not to analyze community consumption of MFAB-bound polysaccharides and its relationship to bacterial growth. Determining the latter is challenging in our animal model; a typical gavage of $10^7$ beads contained 100 µg of bead-immobilized polysaccharide, 1000 times more polysaccharide was consumed daily in the diet by mice in the polysaccharide-supplemented arms of our experiments.

Installing specific functional groups on the surfaces of microscopic paramagnetic glass beads using commercially available organosilane reagents allows 'modular' incorporation of different biomolecules. This approach represents an alternative to a procedure we described recently, where bifunctional biotinylated ligands are generated prior to immobilization on glass beads coated with streptavidin (*Patnode et al., 2019*). Each of these approaches has distinct advantages and disadvantages. By immobilizing ligand directly on the bead surface, MFABs possess a greater number of sites for ligand attachment than streptavidin-coated beads and include no protein constituents. Higher ligand density enables higher levels of ligand loading, which reduces the absolute number of beads that need to be administered, recovered, and processed, and increases the dynamic range of a functional activity readout.

A unifying question for food science, microbiome science, and nutrition research is how to decipher the effects of a nutrient and the matrix in which it resides within a food on modulation of microbiota functions (*Gidley and Yakubov, 2019*; *Leitch et al., 2007*). Crude plant fibers contain various polysaccharides densely intercalated within a cellulose-lignin matrix (*Caffall and Mohnen, 2009*). We found that the absolute abundances of *B. cellulosilyticus* WH2 and *Ruminococcaceae* sp. TSDC 17.2 increase significantly when animals were consuming the crude pea fiber-supplemented diet, but not during supplementation with either of the isolated arabinans. The chemistry for covalent polysaccharide attachment to MFABs not only allows for dense ligand presentation, but also enables multiple ligands to be simultaneously immobilized to create 'hybrid' beads to model the effects of physical colocalization of different fiber components on microbial degradation. In principle, a wide range of different glycan combinations with varying stoichiometries can be explored owing to the fact that different hybrid bead types, each with its own fluorophore, can be created and tested

simultaneously in vitro and in vivo to investigate the mechanism, generality, and biological significance of colocalization of glycans on their degradation by gut microbes.

The selection of gut bacterial taxa for in vivo tests of fiber effects in gnotobiotic animal models is critical for yielding ecologically and physiologically relevant results. Characterizing defined communities comprised of organisms beyond those represented in the 14-member consortium will undoubtedly be informative. For example, members of Lactobacillus, Bifidobacterium, and Faecalibacterium are more tolerant of acidic conditions after carbohydrate fermentation and can outcompete acid intolerant bacteria such as Bacteroides; their inclusion would likely change the response of community members to the presence of different plant fiber preparations and the dynamics of plant fiber degradation documented by various collections of MFABs.

The approach we describe for ligand immobilization does not require the synthesis of bifunctional ligands (or fluorophores); instead, custom functional groups can be incorporated into the probe through modification of the organosilane donor molecule. As such, the MFAB platform provides an opportunity to develop chemistries for nondestructively releasing ligands for analysis (*Bielski and Witczak, 2013*). For example, characterizing microbial degradation of polysaccharides needs to move beyond relatively 'simple' GC–MS measurements of monosaccharides released from the surface of recovered beads to readouts of glycan structures recovered from the bead surface (prior to and after exposure to microbes). This information would provide a more informed view of functional properties (saccharolytic activities) expressed by model communities comprised of cultured sequenced members of a microbiota representing a population of interest, or their intact uncultured microbiota, as well as greater insights about structure/activity relationships of existing or new candidate prebiotic and synbiotic formulations.

# Materials and methods

## Isolation of PFABN

### Fractionation of pea fiber

Raw pea fiber (Rattenmaier) was fractionated using serial extractions with aqueous buffers of increasing harshness (*Pattathil et al., 2012*). Pea fiber (5 g) was defatted by stirring at 23°C for 2 hr in 60 mL of 80% (vol:vol) ethanol. Fiber was pelleted by centrifugation (3500 x *g*, 5 min), and the supernatant was removed. Neat ethanol was added to the pelleted fiber, and the solution was mixed for 2 min. Fiber was centrifuged (3500 x *g*, 5 min), and the supernatant was removed. Neat acetone was added to the pelleted fiber, the solution was mixed for 2 min, centrifuged (3500 x *g*, 10 min), and the supernatant was removed. The resulting 'defatted' pea fiber was dried in a chemical hood overnight. Defatted pea fiber was subsequently resuspended in 200 mL of 50 mM ammonium oxalate (pH = 5.7) and stirred at 23°C for 20 hr. The suspension was centrifuged (7000 x *g*, 15 min), the supernatant was collected, concentrated (Amicon Stirred Cell concentrator [Millipore Sigma; Cat. No.: UFSC20001] with a 3 kDa molecular weight cut-off ultrafiltration disk [Millipore Sigma; Cat. No.: PLBC06210]), and then dialyzed extensively against water (3.5 kDa molecular weight cut-off dialysis tubing [Thermo Scientific; Cat. No.: 88244] or 3.5 kDa molecular weight cut-off Slide-A-Lyzer dialysis cassettes [Thermo Scientific]). The precipitate from the dialysis was recovered by centrifugation (15,000 x *g*, 15 min). The precipitate and soluble material from the dialysis, representing fractions one and two, respectively, were dried with lyophilization.

The pellet from the ammonium oxalate extraction was washed with 200 mL of water, centrifuged (4000 x *g*, 15 min), and the supernatant was discarded. The pellet was resuspended in 200 mL of 50 mM sodium carbonate (pH 10) containing 0.5% (wt:wt) sodium borohydride and stirred at 23°C for 20 hr. The suspension was centrifuged (6000 x *g*, 15 min), and the supernatant was collected. Borohydride was quenched by slowly adding glacial acetic acid. A stringy precipitate began to form as the pH decreased. The suspension was concentrated (as above); the insoluble and soluble portions of the resulting concentrated sodium carbonate suspension were separated with centrifugation (15,000 x *g*, 15 min), yielding fractions three and four, respectively. Fractions were dialyzed and dried with lyophilization.

The pellet from the sodium carbonate extraction was washed with water before resuspension in 200 mL of 1 M potassium hydroxide containing 1% wt:wt sodium borohydride and stirring for 20 hr at 23°C. The suspension was centrifuged (6000 x *g*, 15 min), and the supernatant was removed. Five

drops of 1-octanol were added to prevent foaming during borohydride quenching. A light precipitate began to form in the solution as the pH decreased. The suspension was concentrated; the insoluble and soluble portions of the concentrated 1 M potassium hydroxide extract were separated by centrifugation (15,000 x *g*, 15 min), yielding fractions five and six, respectively. The fractions were dialyzed and dried with lyophilization.

The pellet from the 1 M potassium hydroxide extraction was washed with water before resuspension in 200 mL of 4 M potassium hydroxide containing 1% wt:wt sodium borohydride. The mixture was stirred at 23°C for 20 hr. The suspension was then centrifuged (6000 x *g*, 15 min), and the supernatant was removed. 1-Octanol was added to prevent foaming during borohydride quenching; during this process, a precipitate formed, dissolved, and then reformed as the pH was lowered to 6.0. The resulting suspension was concentrated; the insoluble and soluble portions of the concentrated 4 M potassium hydroxide extract were separated by centrifugation (15,000 x *g*, 15 min), yielding fractions seven and eight, respectively. These fractions were dialyzed and dried with lyophilization. Note that after each extraction, sodium azide was added to a final concentration of 0.05% prior to concentration and dialysis.

## Characterization of pea fiber fractions

Each of the eight fractions was resuspended in water (1 mg/mL) by heating to 90°C and sonication (Branson Sonifer). Insoluble material was removed by centrifugation (18,000 x *g*, 5 min). The soluble material was assayed for protein content (bicinchoninic acid assay) using bovine serum albumin as a standard, DNA content (UV-visible absorbance spectroscopy, Denovix DS-11 spectrophotometer) and total carbohydrate content (phenol-sulfuric acid assay; *Masuko et al., 2005*) using D-glucose as a standard (*Supplementary file 1*). The molecular size of each fraction was measured using an Agilent 1260 high-performance liquid chromatography system equipped with an evaporative light scattering detector. An Agilent Bio Sec-5 column (Cat. No.: 5190–2526) and guard were used with water as the mobile phase. Unbranched pullulan fractions were employed as length standards (Shodex). The monosaccharide composition of each fraction was measured using polysaccharide methanolysis followed by GC–MS (*Doco et al., 2001*). [1,2,3,4,5,6-$^2$H]-*Myo*-inositol (CDN Isotopes) was used as an internal standard. Twofold dilutions of free monosaccharide standards (L-arabinose, D-galactose, D-galacturonic acid, D-glucose, D-glucuronic acid, D-mannose, D-rhamnose, and D-xylose) were simultaneously derivatized and used to quantify the absolute abundance of each monosaccharide in each fraction. GC–MS peaks were quantified using metaMS (*Wehrens et al., 2014*). Glycosyl-linkage analysis was performed on fractions five, seven, and eight at the Complex Carbohydrate Research Center (University of Georgia), employing previously described methods (*Anumula and Taylor, 1992*). Fraction eight was enriched in arabinan and designated PFABN.

## Procedure for scaled up isolation of PFABN

The isolation procedure described above was slightly modified to recover gram quantities of PFABN. Raw pea fiber was resuspended at 50 mg/mL in 1 M potassium hydroxide containing 0.5% (wt:wt) sodium borohydride and stirred at room temperature for 24 hr. The suspension was centrifuged (3900 x *g*, 20 min), and the supernatant was discarded. The pellet from the 1 M potassium hydroxide extraction was resuspended in 4 M potassium hydroxide containing 0.5% (wt:wt) sodium borohydride (50 mg/mL) and stirred at room temperature for 24 hr. The suspension was centrifuged, and the supernatant was collected and neutralized with 4 M acetic acid. Neat ethanol was added (7.5:1 [vol:vol]), and polysaccharide was precipitated at −20°C. Precipitated polysaccharide was isolated by centrifugation (3900 x *g*, 20 min) and rinsed with 250 mL of 80% ethanol (4°C) three times. The pellet was dried overnight under a dry nitrogen stream. The entire procedure was repeated five times to isolate 51 g of PFABN (overall yield 22%). Isolated PFABN was pulverized (Spex SamplePrep Freezer/Mill; Metuchen, NJ; Model 6870), and total carbohydrate content was defined (phenol-sulfuric acid assay).

## GC–MS of neutral monosaccharide composition

PFABN was suspended in water at a concentration of 1 mg/mL and transferred to 8 mm crimp top glass vials (Fisher Scientific; Cat. No.: C4008-632C). One hundred and seventy-five microliters of 2 M trifluoroacetic acid containing 15 ng of D6-*myo*-inositol was added, and the vials were capped with

Teflon-coated aluminum caps (Fisher Scientific; Cat. No.: C4008-2A). PFABN was hydrolyzed for 2 hr at 95°C. Samples were then centrifuged (3200 x $g$, 5 min), the supernatant was transferred to a new glass vial, and the material was dried under reduced pressure. Samples were subsequently oximated by adding 20 µL of methoxyamine (15 mg/mL in pyridine) and incubating the solution overnight at 37°C. Twenty microliters of MSTFA (N-methyl-N-trimethylsilyltrifluoroacetamide plus 1% TCMS [2,2,2-trifluoro-N-methyl-N-(trimethylsilyl)-acetamide, chlorotrimethylsilane]) (Thermo Scientific) was added, and the solution was incubated at 70°C for 1 hr. The material was subsequently diluted with 20 µL heptane before analysis using an Agilent 7890A gas chromatography system coupled with an Agilent 5975C mass spectrometer detector. Employing L-arabinose, D-galactose, D-glucose, D-mannose, D-rhamnose, D-xylose standards, peaks were identified and quantified using metaMS (*Wehrens et al., 2014*); peak areas were corrected using a D6-*myo*-inositol internal standard and quantified using linear fits of twofold diluted standards. Technical replicates represent independent GC–MS derivatizations and analyses of the same sample.

## PFABN linkage analysis

PFABN was enzymatically destarched using amyloglucosidase and α-amylase (Megazyme). To do so, PFABN was first resuspended by heating at 95°C in a solution containing 50 mM sodium malate (pH = 6) and 2 mM calcium chloride (5 mg/mL). Based on the manufacturer's measurement of the specific activities of these two enzymes, we added an amount sufficient to degrade all starch within the PFABN fraction within one minute; nonetheless, we allowed degradation to proceed for 4 hr at 37°C before terminating the reaction by incubation at 95°C for 20 min. Polysaccharide was dialyzed extensively against water and dried by lyophilization. Complete digestion of starch was confirmed with GC–MS analysis of neutral monosaccharides.

Glycosyl-linkage analysis was performed on the destarched PFABN at the Complex Carbohydrate Research Center (University of Georgia) using previously described methods (*Anumula and Taylor, 1992*). Briefly, polysaccharide (1 mg) was taken up in dimethyl sulfoxide, permethylated in the presence of NaOH base, hydrolyzed for 2 hr in 2 M trifluoroacetic acid at 121°C, reduced overnight with sodium borohydride, and acetylated with acetic anhydride and pyridine. Inositol was used as an internal standard. The resulting partially methylated alditol acetates were analyzed by GC–MS (HP-5890 instrument interfaced with a 5970 mass selective detector using a SP2330 capillary column [30 × 0.25 mm ID, Supelco] and a temperature program of 60°C for 1 min, increasing to 170°C at 27.5°C/min, to 235°C at 4°C/min with a 2 min hold, and finally to 240°C at 3°C/min with 12 min hold). Sugar beet arabinan (Megazyme) was analyzed simultaneously. The resulting linkage data are presented in *Supplementary file 1*.

## In vitro growth assays

Bacterial stocks, previously stored at −80°C, were struck onto Brain-heart infusion (BHI; Becton Dickinson) agar plates supplemented with 10% (vol:vol) horse blood. Plates were incubated in an anaerobic growth chamber (Coy Laboratory Products; atmosphere 3% hydrogen, 20% $CO_2$, and 77% $N_2$). Single colonies were picked and grown overnight on a defined Bacteroides minimal medium (BMM) (*McNulty et al., 2013*) containing 5 mg/mL D-glucose. Bacteria were then diluted 1:500 (vol:vol) into BMM supplemented with a carbon source at a final concentration of 0.5% (wt:wt) and distributed into the wells of a 96-well half-area plate (Costar; Cat. No.: 3696). Plates were sealed with an optically clear membrane (Axygen; Cat. No.: UC500) and growth at 37°C was monitored by measuring optical density at 600 nm every 15 min (Biotek Eon instrument with a BioStack 4). Carbon sources tested include D-glucose, PFABN, SBABN, and glucomannan (Megazyme). All conditions were tested in quadruplicate. Readings obtained from control wells inoculated with bacteria but lacking a carbon source were averaged and subtracted from data obtained from carbon-supplemented cultures to generate background subtracted $OD_{600}$ growth curves.

## Gnotobiotic mouse experiments

All experiments involving mice were carried out in accordance with protocols approved by the Animal Studies Committee of Washington University in Saint Louis.

## Colonization

Germ-free male C57BL/6J mice were maintained within flexible plastic isolators under a strict 12 hr light cycle (lights on a 0600) and fed an autoclavable mouse chow (Envigo). Animals were colonized with a 14-member microbial community of cultured, sequenced bacterial strains composed of a mixture of type strains or their Tn mutant library equivalent (*Hibberd et al., 2017*; *Wu et al., 2015*) and strains isolated from the lean co-twin of an obesity discordant twin pair (Twin Pair one in *Ridaura et al., 2013*). Bacterial strains were grown to early stationary phase in gut microbiota medium (*Goodman et al., 2011*) or LYBHI medium (*Sokol et al., 2008*). Monocultures were stored at −80°C after addition of an equal volume of PBS (pH 7.4) supplemented with 30% glycerol (vol: vol). Gavage pools were prepared ($2 \times 10^6$ CFUs per strain; equal volumes of each INSeq library) and introduced into mice using a plastic tipped oral gavage needle. Animals receiving communities with Tn mutant libraries were individually housed in cages containing cardboard shelters (for environmental enrichment).

Five days prior to colonization, mice were switched to a HiSF-LoFV diet. This diet was produced using human foods as described in a previous publication (*Ridaura et al., 2013*), freeze-dried, and milled (D90 particle size 980 μm). The milled diet and each of the three diet supplements were weighed and transferred (separately) into sterile screw top containers (Fisher Scientific; Cat. No.: 22-150-244). Diets were sterilized by gamma irradiation (20–50 kilogreys, Steris, Mentor, OH). Sterility was confirmed by culturing material in TYG medium under aerobic and anaerobic conditions. The HiSF-LoFV diet and supplement were combined after transfer into gnotobiotic isolators (raw pea fiber at 10% [wt:wt]; PFABN at 2% [wt:wt], and SBABN at 2% [wt:wt]). Diets were mixed into a paste after adding sterile water (15 mL/30 g of diet). The paste was pressed into a small plastic tray and placed on the floor of the cage. Fresh diet was introduced every 2 days and in sufficient quantity to allow access ad libitum. Autoclaved bedding (Aspen wood chips; Northeastern Products) was changed at least weekly and immediately following a diet switch. Germ-free mice were randomized into groups to match the starting average weights of animals in all treatment groups.

## Community profiling by sequencing

DNA was isolated from fecal samples by bead beading with 250 μL 0.1 mm zirconia/silica beads and one 3.97 mm steel ball in 500 μL of 2× buffer A (200 mM Tris, 200 mM NaCl, 20 mM EDTA), 210 μL 20% (wt:wt) sodium dodecyl sulfate, and 500 μL of phenol:chloroform:amyl alcohol (pH 7.9; 25:24:1) for 4 min. Four hundred and twenty microliters of the aqueous phase was removed; DNA was purified (QIAquick 96 PCR purification kit; Qiagen) according to the manufacture's protocol and eluted into 10 mM Tris–HCl (pH 8.5). Sequencing libraries were prepared from purified DNA by tagmentation with the Nextera DNA Library Prep Kit (Illumina) and custom barcoded primers (*Adey et al., 2010*). Libraries were sequenced (Illumina Nextseq instrument, 75-nt unidirectional reads) to a depth $\geq 1 \times 10^6$ reads per sample. Reads were demultiplexed and mapped to community member bacterial genomes, 2 'spiked-in' bacterial genomes for absolute abundance calculation (see below), and two 'distractor' genomes (*Faecalibacterium prausnitzii*; GenBank assembly accession: GCA_902167865.1; *Bifidobacterium longum* subsp. *infantis*; GenBank assembly accession: GCA_902167615.1; *Raman et al., 2019*), using custom Perl scripts adapted to use Bowtie 2 (*Langmead and Salzberg, 2012*; *McNulty et al., 2011*).

To calculate bacterial absolute abundance, an aliquot containing a known number of two bacteria strains not encountered in mammalian gut communities or in the diet was added to each fecal sample prior to DNA extraction (*Stämmler et al., 2016*) (30 μL of a $2.22 \times 10^8$ cells/mL suspension of *Alicyclobacillus acidiphilus* DSM 14558 [GenBank assembly accession: GCA_001544355.1] and 30 μL of a $9.93 \times 10^8$ cells/mL suspension of *Agrobacterium radiobacter* DSM 30147 [GenBank assembly accession: GCA_000421945.1]; *Wolf et al., 2019*). Community profiling by sequencing (COPRO-Seq) provides an output counts table that is normalized to the informative genome size of each bacterial genome; this is used to generate a normalized relative abundance table. The calculated relative abundances of the spike-in genomes were 0.40 ± 0.19% and 0.29% ± 0.16 (mean ± s.d.), respectively. For a given taxa $i$, in sample $j$, the absolute abundance in genome equivalents per gram of feces was calculated using the normalized relative abundance and the *A. acidiphilus* spike-in (A. a):

$$taxa_{i,j} = \frac{rel\,abundance_{i,j}}{rel\,abundance\,A.a_j} \times \frac{A.a\,cells\,added\,to\,sample_j}{sample\,mass\,(g)_j}$$

To identify bacterial taxa that respond to each diet treatment, absolute abundance data from fecal samples collected after diet supplementation were fit using a linear mixed-effects model (lme4 package; *Bates et al., 2015*). The dependence of bacterial abundance on 'diet by day' was tested. 'Animal' was included as a random variable. Tukey's HSD p-values from the linear models were corrected for multiple hypotheses (*Benjamini and Hochberg, 1995*). Estimated marginal means were calculated from linear models (emmeans package) of absolute abundances for each diet group. To simplify visualization of the effects of each diet supplement, estimated marginal mean values were expressed as a ratio of the marginal mean of all mice prior to the diet switch on dpg 2. Diet-responsive bacterial strains were defined as those whose absolute abundance was significantly different ($p < 0.01$, linear mixed-effects model [Gaussian]; two-way ANOVA with Tukey's HSD, FDR corrected) in $\geq 3$ of the six total diet comparisons (i.e., (1) HiSF-LoFV vs pea fiber, (2) HiSF-LoFV vs PFABN, (3) HiSF-LoFV vs SBABN, (4) pea fiber vs PFABN, (5) pea fiber vs SBABN, or (6) PFABN vs SBABN), and the estimated marginal mean of the diet effect was greater than 1.5 for at least one diet-supplemented group.

## Tn insertion site sequencing

Multi-taxon INSeq (*Wu et al., 2015*) was used to simultaneously measure genetic fitness determinants in five Bacteroides sp. (four of which were identified as fiber responsive). Briefly, MmeI digestion cleaves genomic DNA at a site 20–21 bp distal to the restriction enzyme's recognition sequence in the *mariner* transposon vector. This flanking genomic DNA, and a taxon-specific barcode inserted into the transposon, allow quantitation of each unique insertion mutant member of a given Bacteroides INSeq library.

Purified fecal DNA was processed as previously described (*Wu et al., 2015*). Genomic DNA was digested with MmeI, size selected, ligated to sample-specific adapter primers, size selected, amplified by PCR, and a specific 131 bp final product isolated from a 4% (wt:wt) MetaPhore (Lonza) DNA gel. Purified DNA was sequenced, unidirectionally, on an Illumina HiSeq 2500 platform (50-nt reads) using a custom primer that captures the species-specific barcode. Quantitation of each insertion mutant's abundance (read counts) was determined using custom software (*Wu et al., 2015*). Count data were normalized for library depth (within the same species), a pseudo-count of 8 was added, and the data were $\log_2$ transformed. Transformed count data from dpg 2 and dpg six were used to build linear models (limma package; *Ritchie et al., 2015*) to identify diet supplement-specific genes that significantly altered bacterial abundance (relative to unsupplemented HiSF-LoFV diet). p-values from the linear models were corrected for multiple hypotheses with the Benjamini–Hochberg method.

## Metaproteomic analysis

The protocol for mass spectrometry-based metaproteomic analysis of fecal samples has been described in detail in *Patnode et al., 2019*. Briefly, samples were processed using bead beating and 4% (wt:wt) sodium dodecyl sulfate (SDS) buffer for lysis and solubilization followed by on-filter protease digestion of proteins. SDS is widely used as a detergent in protein extraction protocols for its increased efficiency of whole-cell protein solubilization. A number of proteome publications have demonstrated that inclusion of this detergent in protein extraction protocols increases peptide identification rates from a wide range of proteins, including peptides found in membrane-associated proteins (*Tanca et al., 2013*). We have also documented an increase in total peptide identifications, including membrane protein representation, after inclusion of SDS in the extraction method that we have applied to a variety of microbial systems (*Blakeley-Ruiz et al., 2019*; *Nickels et al., 2020*; *Salvachúa et al., 2020*). While there may still be underrepresentation of some proteins, such as membrane proteins with multiple transmembrane helices, if there is an underrepresentation of proteins encoded by the PULs due to methodology-induced bias, this bias will be inherent and consistent across all samples. As proteins encoded by the PULs are compared across treatment, we reasoned that underrepresentation of these proteins would be the same across all samples and not impact our overall findings.

Only data from peptides that uniquely map to a single protein were considered for analysis. Summed peptide abundance data for each protein was $\log_2$-transformed. Missing data was imputed to simulate 'instrument limit of detection' by calculating the mean and standard deviation of the protein abundance distribution for each sample. Proteins included in this distribution were detected in more than three mice within a given treatment group. Missing values were imputed as mean minus 2.2 times the standard deviation with a width equal to 0.3 times the standard deviation. For species where greater than 100 proteins were quantified, data were normalized with cyclic loess normalization (limma package). Loess-normalized protein abundance data were then used to build linear models (limma package) to identify diet-supplement-responsive proteins (relative to levels in control mice receiving the unsupplemented HiSF-LoFV diet) at dpg 6. p-values from the linear models were corrected for multiple hypotheses.

## Generation of microbiota functional activity biosensors

### Synthesis of amine plus phosphonate functionalized beads

Paramagnetic, 10 µm diameter glass beads (Millipore Sigma) were incubated at 23°C overnight in a solution of 20 mM HEPES (pH 7.4) and 100 mM NaCl. Equal molar amounts of (3-aminopropyl)triethoxysilane (APTS; Sigma–Aldrich) and 3-(trihydroxysilyl)propylmethylphosphonate (THPMP; Sigma–Aldrich) were subsequently added to a suspension of hydrolyzed NHS ester-activated beads in deionized water (*Bagwe et al., 2006*; *Soto-Cantu et al., 2012*). Beads were derivatized at a density of $5 \times 10^6$/mL, and the organosilane reagents were included at 1000-fold excess of what would be required to coat the bead surface (based on four silane molecules per nm², *Soto-Cantu et al., 2012*). The reaction was allowed to proceed for 5 hr at 50°C with shaking and then terminated with three cycles of washing in water (using a magnet to recover the beads after each wash cycle). Beads were stored at 4°C in a sterile solution of 20 mM HEPES (pH 7.2) and 100 mM NaCl at 4°C.

### Amine plus phosphonate bead acetylation

Beads were washed repeatedly with multiple solvents with the goal of resuspending the beads in anhydrous methanol; to do so, beads were washed in water, then methanol, and then anhydrous methanol (1 vol equivalent; $5 \times 10^6$ beads /mL). Pyridine (0.5 vol equivalents) was then added as a base followed by acetic anhydride (0.5 vol equivalents). The reaction was allowed to proceed for 3 hr at 22°C and then terminated by repeated washing in water. Beads were stored in 20 mM HEPES (pH 7.2) and 100 mM NaCl at 4°C.

### Fluorophore labeling of amine plus phosphonate beads

Beads were labeled with the following *N*-hydroxysuccinimide (NHS) ester-activated fluorophores: (1) Alexa Fluor 488 NHS ester (Fisher Scientific), (2) Promofluor 415 NHS ester (PromoKine); (3) Promofluor 633P NHS ester (PromoKine), and (4) Promofluor 510-LSS NHS ester (PromoKine). NHS-activated fluorophores were dissolved in dimethyl sulfoxide (DMSO) at 1 mM. The stock solution of each fluorophore was diluted in DMSO to 10 µM. The fluorophore was conjugated to amine plus phosphonate beads in 20 mM HEPES (pH 7.2) and 100 mM NaCl ($3 \times 10^6$ beads/mL reaction; final concentration of fluorophore in the reaction, 100 nM). The reaction was allowed to proceed for 50 min at 22°C and then terminated by repeated washing with water. Beads were stored in 20 mM HEPES (pH 7.2) and 100 mM NaCl at 4°C.

### Polysaccharide conjugation to fluorophore-labeled amine plus phosphonate beads

Polysaccharides were resuspended at a concentration of 5 mg/mL in 50 mM HEPES (pH 7.8) using heat and sonication with the exception of maltodextrin (DE 13–17; Sigma–Aldrich), which was resuspended at 50 mg/mL. Triethylamine (TEA, 0.5 equivalent) and 1-cyano-4-dimethylaminopyridinium tetrafluoroborate (CDAP; one eq; Sigma–Aldrich; dissolved in DMSO at a concentration of 50 mg/mL) were added to the polysaccharide solution. The optimal concentration of CDAP for polysaccharide activation, without overactivation and aggregation, was found to be 0.2 mg CDAP/mg of polysaccharide (or roughly 7 moles of hexose per mole of CDAP for a hexose polysaccharide). The polysaccharide/TEA/CDAP solution was mixed for 2 min at 22°C to allow for polysaccharide activation. Fluorophore-labeled amine plus phosphonate beads resuspended in 50 mM HEPES (pH 7.8)

were added to the activated polysaccharide solution, and the reaction was allowed to proceed for 15 hr at 22°C (final polysaccharide concentration typically 3.5 mg/mL). Any aggregated beads were disrupted by gentle sonication. Polysaccharide-conjugated beads were reduced by adding 2-picoline borane (one eq; Sigma–Aldrich) dissolved in DMSO (10% wt:wt) and incubating the mixture for 40 min at 40°C. The reaction was terminated with repeated washing with water. Beads were stored in 20 mM HEPES (pH 7.2) and 100 mM NaCl at 4°C.

Beads were counted using flow cytometry. Typically, 5 µL of a polysaccharide-coated bead solution were added to 200 µL of HNTB (20 mM 4-(2-hydroxyethyl)−1-piperazineethanesulfonic acid [HEPES] [pH 7.4], 100 mM NaCl, 0.01% bovine serum albumin [wt:wt], and 0.01% Tween-20 [wt:wt]) containing CountBright Absolute Counting Beads (Thermo Scientific). Beads were analyzed using flow cytometry on a FACSAriaIII instrument (BD Biosciences).

## Analytical characterization of beads

Bead zeta potential was measured to characterize the extent of modification of the bead surface; zeta potential was determined for beads reacted with organosilane reagents with or without surface amine acetylation. Zeta potential measurements were made on a Malvern ZEN3600 instrument using disposable zeta potential cuvettes (Malvern). Beads were resuspended to a concentration of $5 \times 10^5$/mL in 10 mM HEPES (pH 7.2) passed through a 0.22 µm filter (Millipore) and analyzed in triplicate. Measurements were obtained with the default settings of the instrument, using the refractive index of $SiO_2$ as the material and water as the dispersant.

Bead surface amines were quantified using a ninhydrin-based assay per the manufacturer's instructions (Anaspec) with slight modification. Octylamine (Sigma–Aldrich) in 60% (vol:vol) ethanol: water was used to generate a standard curve. Octylamine or beads resuspended in 60% ethanol were added to glass crimp top vials (1 vol equivalent; 100 µL). One volume equivalent of each of three solutions used for detection of amines was added to the vials. Vials were then sealed and incubated at 95°C for 3 min. The reaction mixture was centrifuged (5000 x $g$, 5 min) to remove the beads, and the absorbance at 570 nm was measured from the resulting supernatant (Biotek Eon). One million amine plus phosphonate beads were analyzed, while three million acetylated amine plus phosphonate beads were required for sufficient signal. Surface amines were calculated from linear regression of the octylamine standard curve.

Biotin binding sites on streptavidin-coated beads (Millipore Sigma) were quantified using an avidin and 4'-hydroxyazobenzene-2-carbocylic acid assay according to the manufacturer's instructions (Fisher Scientific). Briefly, $5 \times 10^6$ beads were incubated for 1 hr at 22°C in 0.25 mL of 0.2 mM biotin dissolved in PBS (7.4). The concentration of biotin remaining was compared to the initial biotin solution to quantify the mols of biotin captured by streptavidin-coated beads.

## Quantification of bead-bound polysaccharide

Polysaccharide degradation from beads was quantified by GC–MS as described in the section 'GC-MS of neutral monosaccharide composition'. Briefly, polysaccharide-coated beads were counted using flow cytometry. Beads for hydrolysis were transferred to a 96-well skirted PCR plate (Multimax; Cat. No.: 2668; $3–7 \times 10^4$ beads/well) and washed three times in water using a magnet. Beads were resuspended in 175 µL of 2M trifluoroacetic acid containing 15 ng of D6-*myo*-inositol as an internal standard and then transferred into 8 mm crimp top glass vials. An aliquot was removed from the vial, and flow cytometry was used to quantify the number of beads that had been transferred to that vial. The quantity of monosaccharide released from a bead was determined from the linear fit of standards divided by the number of beads transferred into the hydrolysis vial. For quantifying relative polysaccharide degradation, the absolute amount of monosaccharide released from the bead surface was divided by the mass of that monosaccharide quantified on input beads (with results expressed as a percentage). Samples were omitted when a insufficient number of beads was recovered for analysis or where an instrument malfunction occurred (i.e., broken vial). Biological replicates represent beads from individual animals, bacterial cultures, or input beads collected from FACS. Technical replicates represent independent GC–MS derivatizations and analyses from beads recovered from the same biological sample.

## Use of polysaccharide-coated MFABs in vitro

Individual glycosyl hydrolases, or combinations of enzymes, were added to PFABN-coated beads ($10^6$ beads/mL in 400 µL of 100 mM sodium acetate [pH 4] plus 0.5% bovine serum albumin). All reactions contained a total of 1 unit of enzyme (per the manufacturer's documentation). Beads were incubated with rotation at 37°C and aliquots were removed after 30 min and 20 hr. Beads were then washed ≥3 times with 20 mM HEPES (7.2) and 50 mM NaCl on a magnetic tube stand, incubated at 80°C for 10 min to inactivate any residual enzyme, and stored in HNTB. The absolute mass of PFABN remaining on the bead surface was determined by using GC–MS as described above.

Single bacterial colonies were picked and grown overnight in BMM (*McNulty et al., 2013*) containing 5 mg/mL D-glucose. Bacteria were diluted 1:250 (vol:vol) into BMM supplemented with either D-glucose or destarched PFABN to a final concentration of 0.5% (wt:wt). Bacterial growth was monitored by measuring optical density at 600 nm every 15 min in a 96-well plate (Biotek Eon). Mid-log phase cells were harvested by centrifugation (4000 x *g*; 5 min) and washed once with BMM without a carbon source. Pelleted cells were resuspended in BMM, without or without added destarched PFABN, and aliquoted into sterile 5 mL snap cap tubes (Eppendorf; Cat. No.: 0030119401). PFABN-coated MFABs were sterilized by washing in 70% ethanol (vol:vol) twice on a magnetic tube stand before resuspension in BMM for at least 1 hr within a Coy chamber (atmosphere 3% hydrogen, 20% $CO_2$, and 77% $N_2$). The concentration of MFABs in the final incubation was $1 \times 10^6$ beads/mL. The bacteria/MFAB mixture, with or without added destarched PFABN, was incubated at 37°C with rotation; aliquots (200 µL) were withdrawn at various time points and the number of colony forming units was determined by 10-fold serial dilution in BMM and plating on LYBHI agar plates. Beads were washed ≥3 times with HNTB on a magnetic tube stand. The absolute mass of PFABN remaining on the bead surface was determined by using GC–MS as described above.

## Gavage and recovery of polysaccharide-coated MFABs from mice

Each bead type was individually sterilized by washing in 70% ethanol (vol:vol) twice on a magnetic tube stand before resuspension in HNTB. A pool of 10–15 × $10^6$ beads (2.5–3.75 × $10^6$ per bead type) in 400 µL of HNTB was prepared for each mouse; a 350 µL aliquot of the pool was introduced by oral gavage; the remaining 50 µL was analyzed as the input beads (see above). Beads were isolated from the cecums of mice 4 hr after gavage or from all fecal pellets that had been collected from a given animal during the 3–6 hr period following gavage. Beads were typically used in mice within 30 days after polysaccharide conjugation.

Recovered beads were resuspended in 10 mL of HNTB by pipetting and subsequently by vortexing. The resulting slurry was passed through a 100 µm nylon filter (Corning; Cat. No.: 352360). Beads were isolated from the suspension by centrifugation (500 x *g*, 5 min) through Percoll Plus (GE Healthcare) in a 50 mL conical tube. Beads were recovered from the bottom of the tube; recovered beads from each animal were distributed into four 1.5 mL sterile tubes and washed at least three times with HNTB on a magnetic tube stand until macroscopic particulate debris from intestinal contents were no longer observed. The material from four tubes were subsequently recombined, and beads were stored in HNTB containing 0.01% (wt:wt) sodium azide at 4°C. The procedure described above typically recovers 20–50% of the input beads from the cecum 4 hr post-gavage.

Bead types were purified by fluorescence-activated sorting (FACSAriaIII; BD Biosciences). Aliquots of input beads were sorted throughout the procedure to quantify and monitor sort yield and purity. Bead purity typically exceeded 98%. Sorted beads were centrifuged (1500 x *g*, 5 min), the supernatant was aspirated, and beads were transferred into a 0.2 mL 96-well skirted PCR plate. Beads were washed with HNTB using a magnetic plate holder and stored at 4°C in HNTB plus 0.01% (wt:wt) sodium azide until analysis. Beads were subjected to acid hydrolysis of the bound polysaccharide, and the amount of liberated neutral monosaccharides was determined by GC–MS. All samples of a given bead type were analyzed in the same GC–MS run; however, the order of analysis of a given bead type recovered from animals representing different treatment groups was randomized. If sufficient beads were available, each bead type from each animal was analyzed up to three times.

## Data availability

COPRO-Seq and INSeq datasets are deposited at the European Nucleotide Archive (ENA) under study accession: PRJEB38095. Proteomic data are available in the MassIVE database under project

number: MSV000085341. COPRO-Seq analysis software can be accessed at https://gitlab.com/hib-berdm/COPRO-Seq and INSeq analysis software at https://github.com/mengwu1002/Multi-taxon_analysis_pipeline; a copy has been archived at swh:1:rev:fac437a7d35ecfd53600ff4dc667563dfb251d25.

## Acknowledgements

We are grateful to David O'Donnell, Maria Karlsson, Justin Serugo, Jiye Cheng, Jessica Forman, and Janaki Guruge for superb technical assistance during various phases of this study. We thank Artur Muszynski and Parastoo Azadi (University of Georgia) for linkage analysis of pea fiber and sugar beet arabinans (performed with support from the Chemical Sciences, Geosciences and Biosciences Division, Office of Basic Energy Sciences, U.S. Department of Energy grant [DE-SC0015662] to DOE-Center for Plant and Microbial Complex Carbohydrates at the Complex Carbohydrate Research Center), and Sanmathi Subbenaik (Nano Research and Environmental Laboratory at Washington University in St. Louis) for her help with zeta potential measurements. Michael Barratt and Michael Patnode provided many helpful suggestions throughout the course of this work. These studies were supported by grants from the NIH (DK70977, DK078669) and Mondelez International. DAW is a Damon Runyon Fellow supported by the Damon Runyon Cancer Research Foundation (DRG–2303–17). ZWB is supported by a National Institutes of Health grant (F30 DK123838). JIG is the recipient of a Thought Leader Award from Agilent Technologies.

## Additional information

### Competing interests

Jeffrey I Gordon: Co-founder of Matatu, Inc., a company characterizing the role of diet-by-microbiota interactions in animal health. A provisional patent on the MFAB technology has been submitted (Washington University, assignee; PCT Application PCT/US2020/042678). The other authors declare that no competing interests exist.

### Funding

| Funder | Grant reference number | Author |
|---|---|---|
| National Institutes of Health | DK70977 | Jeffrey I Gordon |
| National Institutes of Health | DK078669 | Jeffrey I Gordon |
| Damon Runyon Cancer Research Foundation | DRG-2303-17 | Darryl A Wesener |
| National Institutes of Health | F30 DK123838 | Zachary W Beller |
| Agilent Technologies | Thought Leader Award | Jeffrey I Gordon |

The funders had no role in study design, data collection and interpretation, or the decision to submit the work for publication.

### Author contributions

Darryl A Wesener, Conceptualization, Data curation, Formal analysis, Investigation, Methodology, Writing - original draft, Writing - review and editing; Zachary W Beller, Data curation, Formal analysis, Investigation; Samantha L Peters, Data curation, Formal analysis, Investigation, Methodology; Amir Rajabi, Gianluca Dimartino, Investigation; Richard J Giannone, Formal analysis, Investigation, Methodology, Writing - review and editing; Robert L Hettich, Formal analysis, Supervision, Investigation, Writing - review and editing; Jeffrey I Gordon, Conceptualization, Formal analysis, Supervision, Funding acquisition, Methodology, Writing - original draft, Project administration, Writing - review and editing

## Author ORCIDs
Darryl A Wesener (iD) https://orcid.org/0000-0002-8132-7018
Zachary W Beller (iD) https://orcid.org/0000-0003-3965-9147
Samantha L Peters (iD) https://orcid.org/0000-0002-3755-9786
Jeffrey I Gordon (iD) https://orcid.org/0000-0001-8304-3548

## Ethics
Animal experimentation: All experiments involving mice were carried out in accordance with protocols approved by the Animal Studies Committee of Washington University in Saint Louis.

## Decision letter and Author response
Decision letter https://doi.org/10.7554/eLife.64478.sa1
Author response https://doi.org/10.7554/eLife.64478.sa2

# Additional files
## Supplementary files
• Supplementary file 1. Characterization of fractions isolated from pea fiber. (A) Initial procedure that yielded eight fractions during sequential extraction. (B) Glycosyl-linkage analysis of PFABN and SBABN. (C) Summary data of Bacteroides growth in defined minimal medium supplemented with glucose, PFABN, and SBABN.

• Supplementary file 2. Effects of pea fiber, PFABN, and SBABN supplementation of the HiSF-LoFV diet on the absolute abundances of members of the defined community in gnotobiotic mice. (A,B) COPRO-Seq results obtained from treatment groups described in *Figure 2* (mean ± s.d.). (C) COPRO-Seq results obtained from treatment groups described in *Figure 5* (mean ± s.d.).

• Supplementary file 3. Metaproteomic analysis of the effects of pea fiber, PFABN, and SBABN supplementation of the HiSF-LoFV diet on gene expression in members of the defined community. Cyclic loess normalized protein abundance Z-scores of (A) *Bacteroides caccae* TSDC17.2–1.2, (B) *Bacteroides cellulosilyticus* WH2, (C) *Bacteroides finegoldii* TSDC17.2–1.1, (D) *Bacteroides massiliensis* TSDC17.2–1.1, (E) *Bacteroides ovatus* ATCC8483, (F) *Bacteroides thetaiotaomicron* VPI-5482, (G) *Bacteroides vulgatus* ATCC8482, (H) *Collinsella aerofaciens* TSDC17.2–1.1, (I) *Escherichia coli* TSDC17.2–1.2, (J) *Odoribacter splanchnicus* TSDC17.2–1.2, (K) *Parabacteroides distasonis* TSDC17.2–1.1, (L) *Ruminococcaceae* sp TSDC17.2–1.2, (M) *Subdoligranulum variabile* TSDC17.2–1.1.

• Supplementary file 4. INSeq analysis of fitness determinants in diet-responsive *Bacteroides* represented in the defined community as a function of pea fiber, PFABN and SBABN supplementation of the HiSF-LoFV diet. Summary statistics from linear models of gene fitness during (A) pea fiber, (B) PFABN, and (C) SBABN supplementation in (1) *Bacteroides cellulosilyticus* WH2, (2) *Bacteroides ovatus* ATCC8483, (3) *Bacteroides thetaiotaomicron* VPI-5482, and (4) *Bacteroides vulgatus* ATCC8482.

• Supplementary file 5. GC–MS analysis of the mass of monosaccharides bound to the surface of MFABs prior to and after their introduction into gnotobiotic mice (related to *Figures 4* and *5*). (A,B) Results from experiments described in *Figure 4*. (C) Results from experiments depicted in *Figure 5*.

• Transparent reporting form

## Data availability
Data availability: COPRO-Seq and INSeq datasets are deposited at the European Nucleotide Archive (ENA) under study accession PRJEB38095. Proteomic data are available in the MassIVE database under project number MSV000085341. COPRO-Seq analysis software can be accessed at https://gitlab.com/hibberdm/COPRO-Seq, and INSeq analysis software can be accessed at https://github.com/mengwu1002/Multi-taxon_analysis_pipeline (copy archived at https://archive.softwareheritage.org/swh:1:rev:fac437a7d35ecfd53600ff4dc667563dfb251d25/).

The following datasets were generated:

| Author(s) | Year | Dataset title | Dataset URL | Database and Identifier |
|-----------|------|---------------|-------------|-------------------------|
| Wesener DA | 2021 | DNA sequencing data used for COPRO-Seq and INSeq analyses | https://www.ebi.ac.uk/ena/data/search?query=PRJEB38095 | EBI European Nucleotide Archive, PRJEB38095 |
| Wesener DA | 2021 | Fecal metaproteomics data | ftp://massive.ucsd.edu/MSV000085341/ | MassIVE, MSV0000 85341 |
| Hibberd M | 2019 | COPRO-Seq | https://gitlab.com/hibberdm/COPRO-Seq | COPRO-Seq, Gitlab |

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

## Appendix 1

### Supplemental results

Identification of *Bacteroides* PULs whose expression and contribution to fitness are significantly affected in vivo by pea fiber, PFABN, and SFABN

To characterize the response of each Bacteroides to the altered nutritional landscape during diet supplementation, we first conducted a meta-proteomic analysis of PUL expression. Fecal communities were sampled 4 days after initiation of diet supplementation (dpg 6) from mice belonging to the three treatment groups receiving supplemented HiSF-LoFV diets and from control animals consuming the unsupplemented diet. Levels of a given protein were cyclic loess-normalized and expressed as a Z-score relative to the abundances of all quantified proteins in each strain and sample: with this approach differences between these protein abundance values in a given community member in mice fed a supplemented compared to unsupplemented diet are not confounded by differences in the abundances of its host bacterium.

The response of individual proteins to each diet supplement was parameterized using linear models generated in limma (*Ritchie et al., 2015*). In total, the abundances of 354, 154, and 156 proteins were significantly different during diet supplementation with pea fiber, PFABN, or SBABN, respectively (adjusted p-value<0.05, FDR corrected; *Supplementary file 3*); of these proteins, 196, 88, and 89, respectively, were encoded within PULs, loci for polysaccharide biosynthesis, and/or were annotated as carbohydrate active enzymes (CAZymes).

To identify PULs whose expression were altered significantly, we performed gene set enrichment analysis (GSEA) on normalized fecal metaproteomic data, considering those PULs that had at least five encoded proteins whose abundances were altered in the same direction (adjusted p-value<0.05, unpaired one-sample Z-test, FDR corrected; *Luo et al., 2009*). The magnitude of each PUL response was expressed as the average fold-change ($\log_2$) of each quantified protein in that PUL compared to its abundance in the context of the unsupplemented HiSF-LoFV diet. GSEA identified 14, 12, 11, and 8 PULs that we deemed 'responsive' to at least one of the diet supplements in *B. thetaiotaomicron* VPI-5482 (BT), *B. ovatus* ATCC 8483 (Bovatus), *B. cellulosilyticus* WH2 (BcellWH2), and *B. vulgatus* ATCC 8482 (BVU), respectively (*Figure 2—figure supplement 1*).

Plots of fitness score versus change in protein abundance were then generated for all genes in each Bacteroides under each diet condition tested (*Figure 2—source data 1*). High-protein expression and low mutant fitness are represented in the right lower quadrant of these graphs. A chi-square test was used to assess overrepresentation of genes from a given PUL in this quadrant. Genes within a PUL of interest that fell within an ellipse of the interquartile range of both measurements were omitted from the calculation; all genes other than the tested PUL represented the null.

Analyzing PULs as gene sets revealed protein abundance responses and effects on fitness that were shared between the three supplements and others that were unique. For *B. thetaioatomicron* VPI-5482, PUL7 was the most highly upregulated PUL with unfractionated pea fiber or isolated PFABN; it contains multiple glycoside hydrolase (GH) family 43 and 51 members with reported arabinofuranosidase activity (*Figure 2—figure supplement 1*; *Cartmell et al., 2011*). During supplementation with SBABN, *B. thetaioatomicron* VPI-5482 PUL75 was highly upregulated; it specifies multiple GH enzymes, polysaccharide lyases and carbohydrate esterases involved in depolymerizing the repeating disaccharide RGI backbone (α-(1,2)-L-rhamnose-α-(1,4)-D-galacturonic acid) and is induced in vitro by diverse pectic polysaccharides (*Figure 2—figure supplement 1*; *Luis et al., 2018*; *Martens et al., 2011*). BT_PUL7 was identified as having a significant effect on fitness during pea fiber, PFABN, and SBABN supplementation, while BT_PUL75 had a significant effect on fitness only during SBABN supplementation (*Figure 2—source data 1*) (p<0.05, chi-square test, FDR-corrected). Other PULs in *B. thetaioatomicron* VPI-5482 that respond differently based on supplement type include BT_PUL73 (lower in SBABN) and BT_PUL63 (higher in SBABN) (*Figure 2—figure supplement 1*). Previous RNA-Seq studies had indicated that BT_PUL73 is activated by multiple non-arabinan pectic glycans in vitro (*Martens et al., 2011*) and is involved in homogalacturonan degradation (*Luis et al., 2018*). BT_PUL63 responds in vitro to arabinogalactan and modestly to arabinan (*Martens et al., 2011*) and contains enzymes essential for RGI backbone depolymerization (GH145 and GH105), multiple arabinofuranosidases (GH43 and GH51), and α- and β-galactosidases (GH97 and GH35, respectively). Whether BT_PUL63 is responding to the increased mole percent of

galactose, or unique structural features such as 6- or 3,6-substituted galactopyranose in SBABN remains an open question.

*B. vulgatus* ATCC 8482 provided another example of multiple PULs that target arabinan but function as supplement source-specific fitness determinants. BVU_PUL27 and BVU_PUL12 contain genes belonging to GH43, GH51, and GH146 families that have specificity for L-arabinofuranosyl structures found in arabinan (*Figure 2—figure supplement 1B*; *Luis et al., 2018*). BVU_PUL27 expression is responsive to all three supplements (*Figure 2—figure supplement 1A*), but it only significantly affects fitness in the context of unfractionated pea fiber and PFABN supplementation (*Figure 2— source data 1*) ($p < 0.05$, chi-square test, FDR corrected). In contrast, BVU_PUL12 responds to and functions as a fitness determinant only during SBABN supplementation (*Figure 2—figure supplement 1A* and *Figure 2—source data 1*) ($p < 0.05$, chi-square test, FDR corrected).

*B. ovatus* ATCC 8483 PUL97 exhibits significantly greater expression and functions as a fitness PUL with all three supplements (*Figure 2—figure supplement 1* and *Figure 2—source data 1*) ($p < 0.05$, chi-square test, FDR corrected); it is likely involved in RGI utilization based on the presence of multiple polysaccharide lyases, plus GH2, GH28, GH43, and GH105 family enzymes (*Figure 2— figure supplement 1B*). *B. ovatus* ATCC 8483 PUL97 is the only fitness PUL we identified in this strain despite increased expression of other PULs, such as Bovatus_PUL40 and Bovatus_PUL96 during unfractionated pea fiber and PFABN supplementation (*Figure 2—figure supplement 1*).

In *B. cellulosilyticus* WH2, BcellWH2_PUL5 is induced and functions as a fitness PUL in the context of all three supplements (*Figure 2—figure supplement 1* and *Figure 2—source data 1*) ($p < 0.05$, chi-square test, FDR corrected). Expression of BcellWH2_PUL71 is significantly higher only during supplementation with unfractionated pea fiber and PFABN (*Figure 2—figure supplement 1*) but contributes minimally to *B. cellulosilyticus* WH2 fitness (*Figure 2—source data 1*).

Together, these observations illustrate how human gut Bacteroides exhibit great sensitivity to diet supplement source and structure. A substantial number of the genes in our forward genetic screen have a fitness score greater than zero, indicating that the mutant grew better than wild type. The biological explanation for improved fitness in these cases remains to be determined.

**Appendix 1—key resources table**

| Reagent type (species) or resource | Designation | Source or reference | Identifiers | Additional information |
|---|---|---|---|---|
| Strain, strain background (*Bacteroides cellulosilyticus*) | INSeq library (*B. cellulosilyticus* WH2) | *Wu et al., 2015* | | |
| Strain, strain background (*Bacteroides ovatus*) | INSeq library (*B. ovatus* ATCC 8483) | *Wu et al., 2015* | | |
| Strain, strain background (*Bacteroides thetaiotaomicron*) | INSeq library (*B. thetaiotaomicron* 7330) | *Wu et al., 2015* | | |
| Strain, strain background (*Bacteroides thetaiotaomicron*) | INSeq library (*B. thetaiotaomicron* VPI-5482) | *Wu et al., 2015* | | |
| Strain, strain background (*Bacteroides vulgatus*) | INSeq library (*B. vulgatus* ATCC 8482) | *Hibberd et al., 2017* | | |
| Strain, strain background (*Bacteroides cellulosilyticus*) | *B. cellulosilyticus* WH2 | *McNulty et al., 2013* | | |

*Continued on next page*

*Appendix 1—key resources table continued*

| Reagent type (species) or resource | Designation | Source or reference | Identifiers | Additional information |
|---|---|---|---|---|
| Strain, strain background (*Bacteroides ovatus*) | *B. ovatus* ATCC 8483 | ATCC | Cat. No. ATCC 8483 | |
| Strain, strain background (*Bacteroides thetaiotaomicron*) | *B. thetaiotaomicron* 7330 | *Hibberd et al., 2017* | | |
| Strain, strain background (*Bacteroides thetaiotaomicron*) | *B. thetaiotaomicron* VPI-5482 | ATCC | Cat. No. ATCC 29148 | |
| Strain, strain background (*Bacteroides vulgatus*) | *B. vulgatus* ATCC 8482 | ATCC | Cat. No. ATCC 8482 | |
| Strain, strain background (*Bacteroides caccae*) | *B. caccae* TSDC17.2–1.2 | *Ridaura et al., 2013* | | Donor fecal sample F60T2 |
| Strain, strain background (*Bacteroides finegoldii*) | *B. finegoldii* TSDC17.2–1.1 | *Ridaura et al., 2013* | | Donor fecal sample F60T2 |
| Strain, strain background (*Bacteroides massiliensis*) | *B. massiliensis* TSDC17.2–1.1 | *Ridaura et al., 2013* | | Donor fecal sample F60T2 |
| Strain, strain background (*Collinsella aerofaciens*) | *C. aerofaciens* TSDC17.2–1.1 | *Ridaura et al., 2013* | | Donor fecal sample F60T2 |
| Strain, strain background (*Escherichia coli*) | *E. coli* TSDC17.2–1.2 | *Ridaura et al., 2013* | | Donor fecal sample F60T2 |
| Strain, strain background (*Odoribacter splanchnicus*) | *O. splanchnicus* TSDC17.2–1.2 | *Ridaura et al., 2013* | | Donor fecal sample F60T2 |
| Strain, strain background (*Parabacteroides distasonis*) | *P. distasonis* TSDC17.2–1.1 | *Ridaura et al., 2013* | | Donor fecal sample F60T2 |
| Strain, strain background (*Ruminococcaceae* sp.) | *Ruminococcaceae* sp. TSDC17.2–1.2 | *Ridaura et al., 2013* | | Donor fecal sample F60T2 |
| Strain, strain background (*Subdoligranulum variabile*) | *S. variabile* TSDC17.2–1.1 | *Ridaura et al., 2013* | | Donor fecal sample F60T2 |
| Strain, strain background (*Alicyclobacillus acidiphilus*) | *A. acidiphilus* DSM 14558 | DSMZ; *Stämmler et al., 2016* | Cat. No. 14558 | |
| Strain, strain background (*Agrobacterium radiobacter*) | *A. radiobacter* DSM 30147 | DSMZ; *Stämmler et al., 2016* | Cat. No. 30147 | |

*Continued on next page*

*Appendix 1—key resources table continued*

| Reagent type (species) or resource | Designation | Source or reference | Identifiers | Additional information |
|---|---|---|---|---|
| Strain, strain background (*Mus musculus, male*) | C57BL/6J mice; rederived germ-free | The Jackson Laboratory | Cat. No. 00064 | |
| Sequence-based reagent | M12 oligonucleotide, double stranded | *Wu et al., 2015* | | CTGTCCGTTCCGACT ACCCTCCCGAC |
| Sequence-based reagent | INSeq PCR primer; F | *Wu et al., 2015* | | CAAGCAGAAGACGGCA TACG |
| Sequence-based reagent | INSeq PCR primer; R | *Wu et al., 2015* | | AATGA TACGGCGACCACCGAACA CTCTTTCCCTACACGA |
| Sequence-based reagent | INSeq Indexing primer | *Wu et al., 2015* | | ACAGGTTGGATGATAAGT CCCCGGTC |
| Peptide, recombinant protein | Amyloglucosidase | Megazyme | Cat. No. E-AMGFR | |
| Peptide, recombinant protein | alpha-Amylase | Megazyme | Cat. No. E-PANAA | |
| Peptide, recombinant protein | Endo-1,5-α-Arabinanase | Megazyme | Cat. No. E-EARAB | |
| Peptide, recombinant protein | α-L-Arabinofuranosidase (*Aspergillus niger*) | Megazyme | Cat. No. E-AFASE | |
| Peptide, recombinant protein | α-L-Arabinofuranosidase (*Cellvibrio japonicus*) | Megazyme | Cat. No. E-ABFCJ | |
| Peptide, recombinant protein | Endo-Inulinase | Megazyme | Cat. No. E-ENDOIAN | |
| Peptide, recombinant protein | MmeI restriction endonuclease | NEB | Cat. No. R0637L | |
| Peptide, recombinant protein | T4 DNA ligase | NEB | Cat. No. M0202M | |
| Peptide, recombinant protein | Superfi DNA polymerase | Fisher Scientific | Cat. No. 12351050 | |
| Commercial assay or kit | Bicinchoninic acid protein assay kit | Thermo Scientific | Cat. No. 23225 | |
| Commercial assay or kit | Nextera DNA library prep kit | Illumina | Cat. No. 15028211 | |
| Commercial assay or kit | QIAquick 96 PCR purification kit | Qiagen | Cat. No. 28181 | |
| Commercial assay or kit | MinElute gel extraction kit | Qiagen | Cat. No. 28604 | |
| Commercial assay or kit | Quant-iT dsDNA assay kit, high sensitivity | Thermo Scientific | Cat. No. Q33120 | |
| Commercial assay or kit | CountBright absolute counting beads | Thermo Scientific | Cat. No. C36950 | |
| Commercial assay or kit | Ninhydrin test kit | Anaspec | Cat. No. AS-25241 | |
| Commercial assay or kit | Biotin quantitation kit | Thermo Scientific | Cat. No. 28005 | |
| Software, algorithm | R, version 3.5.2 | | https://www.r-project.org/ | |
| Software, algorithm | metaMS | *Wehrens et al., 2014* | | |

*Continued on next page*

*Appendix 1—key resources table continued*

| Reagent type (species) or resource | Designation | Source or reference | Identifiers | Additional information |
|---|---|---|---|---|
| Software, algorithm | COPRO-Seq pipeline | *Hibberd et al., 2017* | https://gitlab.com/hibberdm/COPRO-Seq | |
| Software, algorithm | INSeq pipeline | *Wu et al., 2015* | https://github.com/mengwu1002/Multi-taxon_analysis_pipeline | |
| Software, algorithm | lme4 | *Bates et al., 2015* | https://github.com/lme4/lme4/ | |
| Software, algorithm | emmeans | | https://github.com/rvlenth/emmeans | |
| Software, algorithm | GAGE | *Luo et al., 2009* | | |
| Software, algorithm | limma | *Ritchie et al., 2015* | http://bioconductor.org/packages/release/bioc/html/limma.html | |
| Software, algorithm | FlowJo V10.5.3 | | https://www.flowjo.com/ | |
| Other | Teklad Global 18% Protein Rodent diet | Envigo | Cat. No. 2018S | |
| Other | High saturated fats low fruits and vegetables mouse chow (HiSF-LoFV) | *Ridaura et al., 2013* | | |
| Other | Pea fiber | Rattenmaier | Cat. No. Pea Fiber EF 100 | |
| Other | Sugar beet arabinan | Megazyme | Cat. No. P-ARAB | |
| Other | Glucomannan | Megazyme | Cat. No. P-GLCML | |
| Other | Maltodextrin (DE 13–17) | Sigma–Aldrich | Cat No. 419680 | |
| Other | Gut microbiota medium, for bacterial culture | *Goodman et al., 2011* | | |
| Other | Bacteroides minimal medium, for bacterial culture | *McNulty et al., 2013* | | |
| Other | Pullulan length standards | Shodex | Cat. No. Standard P-82 | |
| Other | [1,2,3,4,5,6-$^2$H]-*Myo*-inositol | CDN Isotopes | Cat. No. D3019 | |
| Other | MSTFA (*N*-methyl-*N*-trimethylsilyltrifluoro)acetamide plus 1% TCMS (2,2,2-trifluoro-*N*-methyl-*N*-(trimethylsilyl)-acetamide, chlorotrimethylsilane) | Thermo Scientific | Cat. No. TS-48915 | |

*Continued on next page*

*Appendix 1—key resources table continued*

| Reagent type (species) or resource | Designation | Source or reference | Identifiers | Additional information |
|---|---|---|---|---|
| Other | PureProteome NHS flexibind magnetic beads | Millipore Sigma | Cat. No. LSKMAGN01 | |
| Other | (3-Aminopropyl) triethoxysilane | Sigma–Aldrich | Cat. No. 440140 | |
| Other | 3-(Trihydroxysilyl) propylmethylphosphonate | Sigma–Aldrich | Cat. No. 435716 | |
| Other | Alexa Fluor 488 NHS ester | Thermo Scientific | Cat. No. A20000 | |
| Other | Promofluor 415 NHS ester | PromoKine | Cat. No. PK-PF415-1-01 | |
| Other | Promofluor 633P NHS ester | PromoKine | Cat. No. PK-PF633P-1–01 | |
| Other | Promofluor 510-LSS NHS ester | PromoKine | Cat. No. PK-PF510LSS-1–01 | |
| Other | 1-Cyano-4-dimethylaminopyridinium tetrafluoroborate | Sigma–Aldrich | Cat. No. RES1458C | |
| Other | 2-Picoline borane | Sigma–Aldrich | Cat. No. 654213 | |
| Other | PureProteome streptavidin magnetic beads | Millipore Sigma | Cat. No. LSKMAGT02 | |
| Other | Percoll Plus | GE Healthcare | Cat. No. 17544502 | |

