## [Decision Letter]

**Acceptance summary:**

Your fluorescent bead technology enables the characterization of specific metabolic features within the gut microbiome. The new methodology has allowed you to begin to explore biochemical activities of gut microbiota in situ. We view this as an important step forward in the area of microbiome research.

**Decision letter after peer review:**

Thank you for sending your article entitled "Microbiota functional activity biosensors for characterizing nutrient utilization in vivo" for peer review at *eLife*. Your article has been evaluated by three peer reviewers, and the evaluation has been overseen by a Reviewing Editor and Wendy Garrett as the Senior Editor.

There was enthusiasm for acceptance of your work but some concern about focus. The reviewers are quite interested about the fluorogenic bead technology and its application for microbiome studies. They have made recommendations about how you might strengthen the connection between the results describing how Bacteroides degrade these fibers and the technology. Consensus of the reviewers is that this is a manuscript about a powerful tool and its application rather than a deep dive into the underlying biology. If you can recast the work in light of the reviewers' comments and concerns and modify the text so that it is more conservative it its claims, we are of the opinion that it would be quite appropriate for publication in *eLife*. The reviewers express a desire to add some minor additional supporting data for some of your experiments (for example as regards Figure 5D). We are providing reviewer comments but to clarify are not asking for extensive new experimentation.

Reviewer #1

I agree with the conclusions the authors draw from the enumeration, INSeq, and proteomic measurements presented in Figure 2 (and its supplements) and Figure 5 - in short, that specific PULs harbored by strains in the synthetic community determine the degree of glycan degradation observed on the beads recovered from mice. It might be nice to run a confirmatory experiment with, e.g., a community in which wild-type BT is replaced by a δ-PUL7 or δ-PUL75 mutant, but the data are already convincing in their current form.

Likely outside the scope of this manuscript, but are the authors able to isolate bacterial cells or genetic material from beads recovered from a mouse? One could imagine an anchor organism that consumes the fiber or simply uses it as a physical niche. Might be especially interesting if bead-associated communities had structure and could be imaged using FISH.

Reviewer #2:

Wesener and colleagues present a method for characterizing biochemical activities of gut microbes in vivo. First, the authors purified and extensively characterized a polysaccharide fraction rich in arabinan (PFABN) and compared the biological activity of PFABN to that of unfractionated pea fiber and arabinan purified from sugar beet (SBABN) using in vitro growth assays/substrate degradation using monocultures of several Bacteroides species, establishing the impact of these on bacterial metabolism. They also characterized responses to these fibers in vivo, in mice colonized with a synthetic community composed of 14 species, consuming diet with/without these fibers. Authors used community-wide proteomic measurements and transposon mutant analyses (TnSeq) for five Bacteroides species to investigate genes involved in the utilization of these fibers and genes important for fitness in these fibers for the 5 Bacteroides species. In the second half of the paper the authors describe methods for generating “Microbiota Functional Activity Biosensors” (MFABs), which consist of glycans covalently-linked to the surface of fluorescent paramagnetic microscopic glass beads. They characterize MFABs and perform validation studies (i.e., quantification of glycan metabolism) in vitro and in gnotobiotic mice. Using unique fluorogenic tags authors analyze simultaneously multiple bead types with different immobilized polysaccharides. Presented results suggest that colocalizing different polysaccharides on the bead surface can have a synergistic effect on their degradation modulated by diet. These newly developed Microbiota Functional Activity Biosensors are a significant contribution to the field as options for measuring gut microbiota biochemical activities in vivo are currently very limited. Also, the study contributes valuable datasets on bacterial (proteomics/InSeq for 5 Bacteroides species) that will be useful to other scientists studying the microbiome. Specific comments:

The paper comes across as having two major parts that could be better integrated. Testing whether any of the key PULs identified for degradation of PFABN/SBABN (results from INSeq/proteomics) impact degradation of these fibers in the beads in vitro (or vivo) would strengthen the connection between the results describing how Bacteroides degrade these fibers and the new development.

Not clear how the statistical differences in growth were determined in Figure 1. i.e., what values were used? Growth rate during exponential phase? Should be indicated.

Bead assays: What fraction of the beads orally-gavaged are recovered from the mice? Also, in the Materials and methods authors mention that beads were collected from cecum and/or from fecal pellets. Does collection site affect the extent to which glycans on the beads are metabolized?

Beads assays ii. The measurement of glycans that remain bound to the beads informs what it is released from them. It is likely that in the context of a complex community most/all? of the released carbohydrates are also used by microbes. This may not always be true, particularly in monocultures or in experiments in mice with small communities. This should be discussed.

"The results disclosed that the fitness scores of 332, 195, and 75 genes were significantly altered during diet supplementation with pea fiber, PFABN, or SBABN, respectively (Supplementary file 4) (adjusted p-value < 0.05, FDR-corrected)" Not clear what species is being discussed here. Supplementary file 4 contains 12 tables (3 supplemented substrates/4 Bacteroides strains).

Reviewer #3:

This manuscript represents a technical tour de force, and describes an approach to trace and understand metabolic pathways generated by single compounds in a complex ecosystem such as the mouse gut. An outstanding research team has applied many state of the art methodologies to examine the effects of particular dietary substrates on bacterial growth and gene expression. This represents a technological advance but the extent of the advance is not clear, since the results often are marginal. Although a great deal of work has been performed, the overall significance of the work is not obvious.

Several specific points should be addressed as well:

The pea fiber arabinan is only 71% of the purified polysaccharide. What is the significance of the substantial carbohydrate contamination? Can it really be called purified with such extensive contamination? The results in Figure 4 indicate that the xylan component materially affects the results.

What is is the significance of the long lag phase for *B. ovatus* with glucose as the substrate? That pattern is unlike all other conditions.

Figure 2. Panels B and C seem to be contradictory. All of the indicated bacteria grew as well on PF as on the “purified” polysaccharides in panels B (and in all cases grew better than on PFABN, but a marked difference is shown in panel C. This is difficult to reconcile.

Figure 2—figure supplement 2. Not surprisingly, results for PF and PFABN were similar, but they were not very different from SBABN. It is difficult to understand what actually was shown here. Although elegant, it is unclear what the fitness calculations add to the calculations of protein abundance. Almost all of the significant variance is along the axis of the protein abundances. A considerable number of the data points for the specified proteins are above the zero line for fitness. (mutant grew better than wild-type). The authors should account for the fitness variation in both directions, and not just cherry-pick the ones that support their hypothesis.

Figure 3C. Why are the results shown for SBABN when he main thrust of the paper is PFABN? If the method is universal, then the PFABN results should be similar.

Figure 4. Panel B is confusing-are the germ-free mice shown under the heading colonized? That is the only way to reconcile the text and figure. But then why are sugars metabolized in the GF animals? Or are the GF mice not shown? This point needs to be clarified since it affects the interpretation of the results.

Figure 5 panel D. The y-axis in the two sub-panels that deal with the glucomannan coated beads are different. In the HISF/LoFV columns, the results of the 2 conditions differ, but the pea fiber results are the same across the panels-this difference is the source of the author's claims, but this seems in the realm of experimental variation, rather than biological meaning. There should be more confirmation of the points that the authors are making, since they end with a strong statement of impact without the necessary rigor.

Discussion: In general, the Discussion overstates the significance of the actual findings. The broad statements should be toned down. The limitations of the purification methods should be acknowledged and how that affects the results achieved should be addressed.

---

## [Author Response]

Reviewer #1I agree with the conclusions the authors draw from the enumeration, INSeq, and proteomic measurements presented in Figure 2 (and its supplements) and Figure 5 - in short, that specific PULs harbored by strains in the synthetic community determine the degree of glycan degradation observed on the beads recovered from mice. It might be nice to run a confirmatory experiment with, e.g., a community in which wild-type BT is replaced by a δ-PUL7 or δ-PUL75 mutant, but the data are already convincing in their current form.

We agree with the reviewer that deletion of an entire multi-kilobase PUL would provide confirmatory evidence of the importance of that PUL for polysaccharide utilization and fitness. We are working to develop CRISPR-based tools for removing specified PULs from *Bacteroides*, but this remains a “work in progress”. Therefore, we feel that targeted deletion of these or other PULs is outside the scope of the current study which focuses on development of MFAB technology and its application for quantifying polysaccharide degradation by a gut microbial community.

Likely outside the scope of this manuscript, but are the authors able to isolate bacterial cells or genetic material from beads recovered from a mouse? One could imagine an anchor organism that consumes the fiber or simply uses it as a physical niche. Might be especially interesting if bead-associated communities had structure and could be imaged using FISH.

We very much appreciate the reviewer’s suggestion and vision. However, quantifying the distribution of each member of our 13-member community within and across different bead-types, within and across different regions of the gut as a function of diet using FISH or other culture-independent methods is technically challenging. We agree with the reviewer at two levels: such an effort is outside the scope of the present work but certainly a future application of the bead technology.

Reviewer #2:Wesener and colleagues present a method for characterizing biochemical activities of gut microbes in vivo. First, the authors purified and extensively characterized a polysaccharide fraction rich in arabinan (PFABN) and compared the biological activity of PFABN to that of unfractionated pea fiber and arabinan purified from sugar beet (SBABN) using in vitro growth assays/substrate degradation using monocultures of several Bacteroides species, establishing the impact of these on bacterial metabolism. They also characterized responses to these fibers in vivo, in mice colonized with a synthetic community composed of 14 species, consuming diet with/without these fibers. Authors used community-wide proteomic measurements and transposon mutant analyses (TnSeq) for five Bacteroides species to investigate genes involved in the utilization of these fibers and genes important for fitness in these fibers for the 5 Bacteroides species. In the second half of the paper the authors describe methods for generating “Microbiota Functional Activity Biosensors” (MFABs), which consist of glycans covalently-linked to the surface of fluorescent paramagnetic microscopic glass beads. They characterize MFABs and perform validation studies (i.e., quantification of glycan metabolism) in vitro and in gnotobiotic mice. Using unique fluorogenic tags authors analyze simultaneously multiple bead types with different immobilized polysaccharides. Presented results suggest that colocalizing different polysaccharides on the bead surface can have a synergistic effect on their degradation modulated by diet. These newly developed Microbiota Functional Activity Biosensors are a significant contribution to the field as options for measuring gut microbiota biochemical activities in vivo are currently very limited. Also, the study contributes valuable datasets on bacterial (proteomics/InSeq for 5 Bacteroides species) that will be useful to other scientists studying the microbiome. Specific comments:The paper comes across as having two major parts that could be better integrated. Testing whether any of the key PULs identified for degradation of PFABN/SBABN (results from INSeq/proteomics) impact degradation of these fibers in the beads in vitro (or vivo) would strengthen the connection between the results describing how Bacteroides degrade these fibers and the new development.

As noted in our response to reviewer 1, we are working to develop CRISPR-based tools for PUL deletions. In the revised Discussion, we have emphasized how application of genetic (e.g. CRISPR-based) tools for single gene or entire PUL deletion, combined with the use of MFABs as a functional readout, could provide new insights about the mechanisms by which community members acquire/degrade nutrients (polysaccharides).

We have modified the text:

“Future studies where MFABs are used in conjunction with genetic tools that enable rapid deletion of genes or entire multi-kilobase loci could provide new insights about the mechanisms by which community members acquire/degrade nutrients (polysaccharides).”

Not clear how the statistical differences in growth were determined in Figure 1. i.e., what values were used? Growth rate during exponential phase? Should be indicated.

We have modified the text to state:

“in vitro assays performed in a minimal defined medium (McNulty et al., 2013) with Bacteroides type strains established that *B. ovatus* ATCC 8483, *B. cellulosilyticus* WH2 and *B. thetaiotaomicron* VPI-5482 grew on isolated PFABN and SBABN, although less rapidly during the exponential phase of growth and to a lower cell density than in medium containing an equivalent concentration of d-glucose (results based on measurements of OD600; see Figure 1C, Supplementary file 1).”

Note, we have provided data summarizing the growth curves in Supplementary file 1 (maximum change in OD600 per hour and maximum OD600 achieved).

Bead assays: What fraction of the beads orally-gavaged are recovered from the mice? Also, in the Materials and methods authors mention that beads were collected from cecum and/or from fecal pellets. Does collection site affect the extent to which glycans on the beads are metabolized?

We have modified the *Materials and methods* section to read:

“The procedure described above typically recovers 20-50% of the input beads from the cecum four hours post gavage.”

Beads from Experiment #1 were recovered from the cecum and beads from Experiment #2 were isolated from the feces. In general, the results obtained were similar; this was reported in the original manuscript:

“[see Figure 4B,C and Supplementary file 5 which provides evidence that results from cecal samples (Experiment 1) and fecal samples (Experiment 2) were comparable].”

Beads assays ii. The measurement of glycans that remain bound to the beads informs what it is released from them. It is likely that in the context of a complex community most/all? of the released carbohydrates are also used by microbes. This may not always be true, particularly in monocultures or in experiments in mice with small communities. This should be discussed.

We agree and have modified the manuscript throughout to emphasize that MFABs report on polysaccharide degradation. For example, the Discussion has been modified to include:

“We emphasize that in this study, we use MFABs to quantify community degradative activity and not to analyze community consumption of MFAB-bound polysaccharides and its relationship to bacterial growth. Determining the latter is challenging in our animal model; a typical gavage of 10^7^ beads contained 100 micrograms of bead immobilized polysaccharide, 1,000 times more polysaccharide was consumed daily in the diet by mice in the polysaccharide-supplemented arms of our experiments.”

"The results disclosed that the fitness scores of 332, 195, and 75 genes were significantly altered during diet supplementation with pea fiber, PFABN, or SBABN, respectively (Supplementary file 4) (adjusted p-value < 0.05, FDR-corrected)" Not clear what species is being discussed here. Supplementary file 4 contains 12 tables (3 supplemented substrates/4 Bacteroides strains).

We have modified the text to link the number of genes to specific *Bacteroides* species:

“The results disclosed that the fitness scores of a total of 39 genes in *B. thetaiotaomicron* VPI-5482, 135 genes in *B. ovatus* ATCC 8483, 346 genes in *B. cellulosilyticus* WH2, and 82 genes in *B. vulgatus* ATCC 8482 were significantly decreased during diet supplementation with either pea fiber, PFABN, or SBABN (Supplementary file 4) (adjusted p-value < 0.05, FDR-corrected).”

Reviewer #3:This manuscript represents a technical tour de force, and describes an approach to trace and understand metabolic pathways generated by single compounds in a complex ecosystem such as the mouse gut. An outstanding research team has applied many state of the art methodologies to examine the effects of particular dietary substrates on bacterial growth and gene expression. This represents a technological advance but the extent of the advance is not clear, since the results often are marginal. Although a great deal of work has been performed, the overall significance of the work is not obvious.Several specific points should be addressed as well:The pea fiber arabinan is only 71% of the purified polysaccharide. What is the significance of the substantial carbohydrate contamination? Can it really be called purified with such extensive contamination? The results in Figure 4 indicate that the xylan component materially affects the results.

Arabinose represents 71 mole percent of the isolated pea fiber arabinan fraction. Similar to arabinan isolated from sugar beet, pea fiber arabinan contains galactose, likely as covalently attached galactan fragments, and glucose as starch (see Figure 1A).

The reviewer is correct: xylose is present in the pea fiber arabinan fraction (12 mole percent). We had described the presence of xylan and starch in the text:

“Xylose was present as a linear xylan (4-substituted xylose) and glucose in the form of residual starch.”.

We have modified the text to include our rationale for naming the polysaccharide pea fiber arabinan and have replaced the word “purified” with “isolated”.

“Xylose was present as a linear xylan polysaccharide (4-substituted xylose) and glucose in the form of residual starch. We named the isolated polysaccharide fraction “pea fiber arabinan” (PFABN) based on (i) these linkage results, (ii) the fact that arabinose comprises the majority of its monosaccharide content (71 mole percent) and (iii) our observation that ~90% of all non-starch carbohydrate is represented by what is likely a single species of polysaccharide, with the remaining being xylan.”

What is the significance of the long lag phase for B. ovatus with glucose as the substrate? That pattern is unlike all other conditions.

We assume the reviewer is referring to *B. vulgatus*. We do not understand why this organism exhibited a long lag phase when grown in defined minimal medium containing glucose as the sole carbon source, as opposed to either of the two isolated arabinan polysaccharides. All monocultures were seeded at the same density from a culture that had been grown overnight in minimal medium supplemented with glucose (Materials and methods).

Figure 2. Panels B and C seem to be contradictory. All of the indicated bacteria grew as well on PF as on the “purified” polysaccharides in panels B (and in all cases grew better than on PFABN, but a marked difference is shown in panel C. This is difficult to reconcile.

We appreciate the reviewer’s confusion. The key point relates to the mass of diet supplement consumed and the mass of arabinan contained in the different types of supplement.

We took care to make sure that the base HiSF-LoFV diet was supplemented with unfractionated pea fiber, isolated PFABN or isolated SBABN at levels such that the *mass of arabinan* consumed each day by mice in the different treatment groups would be equal.

We noted in the original manuscript that the isolated PFABN fraction accounted for ~20% of the dry mass of intact pea fiber. Thus, we made sure that the mass of unfractionated pea fiber in the supplemented HiSF-LoFV diet was 5-times higher than the mass of PFABN or SBABN added. This point is described in Figure 2A and its legend.

To account for these differences in the mass of supplement consumed by mice daily, we used the specific activity-like calculation described in Figure 2C and its legend. Moreover, in addition to panel C, we included panel B in the Figure to show the change in absolute abundance of responsive community members.

To make these points clearer, we have modified the text to state :

“This metric of specific activity, which takes into account the absolute mass of supplement consumed daily by members of the different groups of mice, revealed that both of the isolated arabinan preparations had a significantly greater effect compared to intact pea fiber at the doses tested (Figure 2C) [p<0.05 mixed-effects linear model (Gaussian); one-way ANOVA with Tukey’s honest significant difference, FDR-corrected].”

Figure 2—figure supplement 2. Not surprisingly, results for PF and PFABN were similar, but they were not very different from SBABN. It is difficult to understand what actually was shown here. Although elegant, it is unclear what the fitness calculations add to the calculations of protein abundance. Almost all of the significant variance is along the axis of the protein abundances. A considerable number of the data points for the specified proteins are above the zero line for fitness. (mutant grew better than wild-type). The authors should account for the fitness variation in both directions, and not just cherry-pick the ones that support their hypothesis.

Fitness benefits following gene deletion are observed in forward genetic screens, although the biological meaning of increased fitness can be obscure. We have paraphrased the point raised by the reviewer by noting in our discussion of the INSeq results that:

“A substantial number of the genes in our forward genetic screen have a fitness score greater than zero, indicating that the mutant grew better than wild-type. The biological explanation for improved fitness in these cases remains to be determined.”

Figure 3C. Why are the results shown for SBABN when he main thrust of the paper is PFABN? If the method is universal, then the PFABN results should be similar.

Sugar beet arabinan was used when developing the MFAB conjugation reaction because it is commercially available in large quantities and is similar in composition to pea fiber arabinan.

In our development of the CDAP immobilization chemistry for use with MFABs, we have not yet encountered a carbohydrate-based polymer that is not amenable to conjugation. We provide data using ligands with varying physical features (i.e., chain length, monosaccharide composition). The results yielded similar ligand loading levels (2-20 ng bound per 1,000 beads); this includes our findings with a model oligosaccharide, maltodextrin (see Figure 3—figure supplement 3C).

Figure 4. Panel B is confusing-are the germ-free mice shown under the heading colonized? That is the only way to reconcile the text and figure. But then why are sugars metabolized in the GF animals? Or are the GF mice not shown? This point needs to be clarified since it affects the interpretation of the results.

We are sorry for the confusion. Our results from beads recovered from germ-free mice are not included in main text Figure 4. Rather, the results of quantifying six monosaccharides released from three bead types recovered from germ-free mice are presented, and compared to input beads, in Figure 4—figure supplement 3.

We have modified the text to emphasize that only results from colonized animals are presented in Figure 4:

“In contrast to germ-free controls (see Figure 4—figure supplement 3), the mass of arabinose was significantly decreased when PFABN- or SBABN-coated beads were recovered from colonized mice fed the unsupplemented HiSF-LoFV diet (Figure 4B,C; p<0.05, Mann-Whitney U test).”

We have also modified the title of Figure 4 to include “colonized”.

“Figure 4. Quantifying microbial degradation of PFABN- and SBABN-coated beads in colonized gnotobiotic mice fed unsupplemented or supplemented HiSF-LoFV diets.”

Figure 5 panel D. The y-axis in the two sub-panels that deal with the glucomannan coated beads are different. In the HISF/LoFV columns, the results of the 2 conditions differ, but the pea fiber results are the same across the panels-this difference is the source of the author's claims, but this seems in the realm of experimental variation, rather than biological meaning. There should be more confirmation of the points that the authors are making, since they end with a strong statement of impact without the necessary rigor.

We have modified Figure 5D to make the y-axis equivalent across the subpanels and the text to state:

“These results show that deliberate physical colocalization can result in quantitatively modest, albeit statistically significant synergistic degradation of polysaccharides during fiber supplementation (Supplementary file 5; p<0.05, linear model; diet supplement by bead type interaction term).”

The Discussion now contains the caveat:

“In principle, a wide range of different glycan combinations with varying stoichiometries can be explored owing to the fact that different hybrid bead types, each with its own fluorophore, can be created and tested simultaneously in vitro and in vivo to investigate the mechanism, generality, and biological significance of this observation.”

Discussion: In general, the Discussion overstates the significance of the actual findings. The broad statements should be toned down. The limitations of the purification methods should be acknowledged and how that affects the results achieved should be addressed.

We appreciate the reviewer’s thoughtful comment and agree. We have modified the Discussion in the ways suggested. For example;

We have emphasized the fact that MFABs were used to monitor degradation of polysaccharides not their consumption by members of the community.

We now state in the text that:

“The bead-based Microbiota Functional Activity Biosensors (MFABs) described in this report represent a technology for measuring biochemical activities expressed by a microbial community of interest in vivo or ex vivo. Future studies where MFABs are used in conjunction with genetic tools that enable rapid deletion of genes or entire multi-kilobase loci could provide new insights about the mechanisms by which community members acquire/degrade nutrients (polysaccharides). We emphasize that in this study, we use MFABs to quantify community degradative activity and not to analyze community consumption of MFAB-bound polysaccharides and its relationship to bacterial growth. Determining the latter is challenging in our animal model; a typical gavage of 107 beads contained 100 micrograms of bead immobilized polysaccharide, 1,000 times more polysaccharide was consumed daily in the diet by mice in the polysaccharide-supplemented arms of our experiments.”

We have modified the text to tone down a general statement about how MFABs can help design more nutritious foods:

“The selection of gut bacterial taxa for in vivo tests of fiber effects in gnotobiotic models is critical for yielding ecologically and physiologically relevant results.”

We have mentioned caveats associated with the purity of our PFABN preparation in the Discussion:

“Robust isolation procedures and analytical characterization are required to isolate grams of polysaccharide and test their biological effects in gnotobiotic mice. The approach we employed allowed us to isolate 50 grams of material to ~85% purity with the remaining material comprised of starch and xylan. Starch can be degraded by host enzymes. The xylan present in our PFABN preparation could, in principle, influence the microbiota response. However, there is arabinoxylan already present the base HiSF-LoFV diet and we found that expression of known xylan-responsive PULs in *B. cellulosilyticus* WH2 and *B. ovatus* ATCC 8483 was not induced by PFABN supplementation (Figure 2— figure supplement 1) (Patnode et al., 2019; McNulty et al., 2013; Martens et al., 2011).”